# Drug specificity and affinity are encoded in the probability of cryptic pocket opening in myosin motor domains

Artur Meller[1,2], Jeffrey M Lotthammer[1], Louis G Smith[1,3], Borna Novak[1,2], Lindsey A Lee[4,5], Catherine C Kuhn[1], Lina Greenberg[1], Leslie A Leinwand[4,5], Michael J Greenberg[1], Gregory R Bowman[1,3]*

[1]Department of Biochemistry and Molecular Biophysics, Washington University in St. Louis, St Louis, United States; [2]Medical Scientist Training Program, Washington University in St. Louis, Philadelphia, United States; [3]Department of Biochemistry and Biophysics, University of Pennsylvania, Philadelphia, United States; [4]Molecular, Cellular, and Developmental Biology Department, University of Colorado Boulder, Boulder, United States; [5]BioFrontiers Institute, Boulder, United States

**Abstract** The design of compounds that can discriminate between closely related target proteins remains a central challenge in drug discovery. Specific therapeutics targeting the highly conserved myosin motor family are urgently needed as mutations in at least six of its members cause numerous diseases. Allosteric modulators, like the myosin-II inhibitor blebbistatin, are a promising means to achieve specificity. However, it remains unclear why blebbistatin inhibits myosin-II motors with different potencies given that it binds at a highly conserved pocket that is always closed in blebbistatin-free experimental structures. We hypothesized that the probability of pocket opening is an important determinant of the potency of compounds like blebbistatin. To test this hypothesis, we used Markov state models (MSMs) built from over 2 ms of aggregate molecular dynamics simulations with explicit solvent. We find that blebbistatin's binding pocket readily opens in simulations of blebbistatin-sensitive myosin isoforms. Comparing these conformational ensembles reveals that the probability of pocket opening correctly identifies which isoforms are most sensitive to blebbistatin inhibition and that docking against MSMs quantitatively predicts blebbistatin binding affinities ($R^2$=0.82). In a blind prediction for an isoform (Myh7b) whose blebbistatin sensitivity was unknown, we find good agreement between predicted and measured IC50s (0.67 μM vs. 0.36 μM). Therefore, we expect this framework to be useful for the development of novel specific drugs across numerous protein targets.

## Editor's evaluation

This study presents insights into how conformational dynamics differentially influences drug specificity and affinity in myosin isoforms using computational approaches. The evidence supporting the conclusions is convincing, establishing a relationship between inhibition and protein dynamics using state of the art computational techniques followed by experimental validation. The work is important and will be of broad interest to computational biophysicists and medicinal chemists.

## Introduction

Achieving specificity is a major challenge in the design of novel drugs. An effective drug must bind its target protein tightly and avoid triggering unwanted side effects that might arise due to

*For correspondence: grbowman@seas.upenn.edu

Competing interest: The authors declare that no competing interests exist.

off-target interactions with other proteins. This problem is especially challenging when targeting specific members of protein families when multiple closely related isoforms with similar structures are expressed. Another notoriously difficult problem is targeting enzymes with substrates, such as ATP, that are shared across many protein families (*Longo et al., 2020*), because compounds that compete with endogenous ligands at active sites may trigger off-target effects.

Targeting allosteric sites offers several practical advantages for drug design. Unlike drugs targeting active sites, allosteric compounds can enhance desirable protein functions, in addition to the more classic drug design strategy of inhibiting undesirable activities (*Knoverek et al., 2019*). Allosteric sites are often less conserved than active sites (*Wenthur et al., 2014*), making it easier to achieve specificity. Indeed, several highly specific allosteric compounds have been serendipitously discovered through high-throughput screens. These allosteric compounds target diverse proteins, such as G-protein-coupled receptors (*Dror et al., 2013*), myosins (*Trivedi et al., 2020*), kinases (*Wu et al., 2015*), and β-lactamases (*Hart et al., 2017*). Despite these successes, *de novo*, structure-based, rational drug design efforts targeting allosteric sites are difficult because most experimental structural studies only offer a limited snapshot of a protein's larger conformational landscape. In solution, proteins occupy a diverse set of conformational states, and some allosteric binding sites are not readily apparent from these static structures (*Cimermancic et al., 2016*). Discovering and targeting such 'cryptic' pockets may be an appealing strategy for achieving specificity toward clinically relevant proteins deemed 'undruggable' by conventional structural studies (*Vajda et al., 2018*).

Myosins are a ubiquitous superfamily of ATPases that hold potential as drug targets for numerous diseases. Myosins are responsible for many cellular processes including endocytosis, cell division, muscle contraction, and long-range transport. (*Preller and Manstein, 2012*) Compounds targeting a subset of striated muscle myosins have been developed and shown great promise in clinical trials for heart failure (*Teerlink et al., 2021*) and hypertrophic cardiomyopathy (*Olivotto et al., 2020*). Recently, the myosin inhibitor, mavacamten, received FDA approval for the treatment of symptomatic obstructive hypertrophic cardiomyopathy. (*U.S. Food and Drug Administration, 2022*) Despite this progress, there is a need for additional myosin modulators in the settings of heart and skeletal muscle diseases (*Barrick and Greenberg, 2021*; *Tajsharghi and Oldfors, 2013*), cancer (*Nature Reviews Clinical Oncology, 2014*), and parasitic infections (*Robert-Paganin et al., 2019*). However, targeting specific myosin isoforms remains extremely difficult because there are 38 myosin genes in the human genome with the typical cell in the human body expressing about 20 myosin isoforms. (*Preller and Manstein, 2013*) Furthermore, myosins with divergent cellular roles and biochemical properties share a highly conserved motor domain fold and active site structure. (*Robert-Paganin et al., 2020*) High-throughput screens have revealed a handful of promising small molecules that allosterically inhibit or activate myosins with varying degrees of specificity (*Bond et al., 2013*). Developing a quantitative understanding of how these allosteric modulators achieve specificity would improve our ability to design novel therapeutics targeting specific myosin isoforms.

Blebbistatin is a myosin-II specific allosteric inhibitor which can be used to understand the molecular mechanisms governing specificity. Blebbistatin was discovered in a high-throughput screen targeting nonmuscle myosin IIs (*Straight et al., 2003*). However, further experiments revealed that blebbistatin broadly inhibits other myosins-II isoforms, such as fast skeletal, β-cardiac, and smooth muscle myosin with varying IC50s, but does not inhibit other myosin families, such as myosin-Xs and myosin-Vs (*Limouze et al., 2004*). Blebbistatin inhibits myosin ATPase by preventing the release of phosphate from the active site and interfering with actin binding (*Kovács et al., 2004*). However, experimental structures of blebbistatin bound to myosin reveal that it binds in a cleft approximately 9 Å from the active site, consistent with its designation as an allosteric effector (*Allingham et al., 2005*).

The mechanism by which blebbistatin differentially inhibits myosin isoforms is not completely understood. Previous studies have posited that blebbistatin specifically inhibits myosin-IIs because of a single residue difference between myosin-IIs and other myosins at the blebbistatin binding site (*Figure 1B*, *Figure 1D*; *Rauscher et al., 2018*). However, it is much less clear what molecular determinants explain differences in blebbistatin potency between isoforms in the myosin-II family. For example, across multiple experiments (*Limouze et al., 2004*; *Eddinger et al., 2007*; *Wang et al., 2008*; *Zhang et al., 2017*; *Várkuti et al., 2016*; *Radnai, 2021*), smooth muscle myosin is inhibited less potently than nonmuscle myosin IIA, despite perfect sequence identity between the residues that line the blebbistatin-binding pocket in these two isoforms (*Figure 1B*, *Supplementary file 1A*).

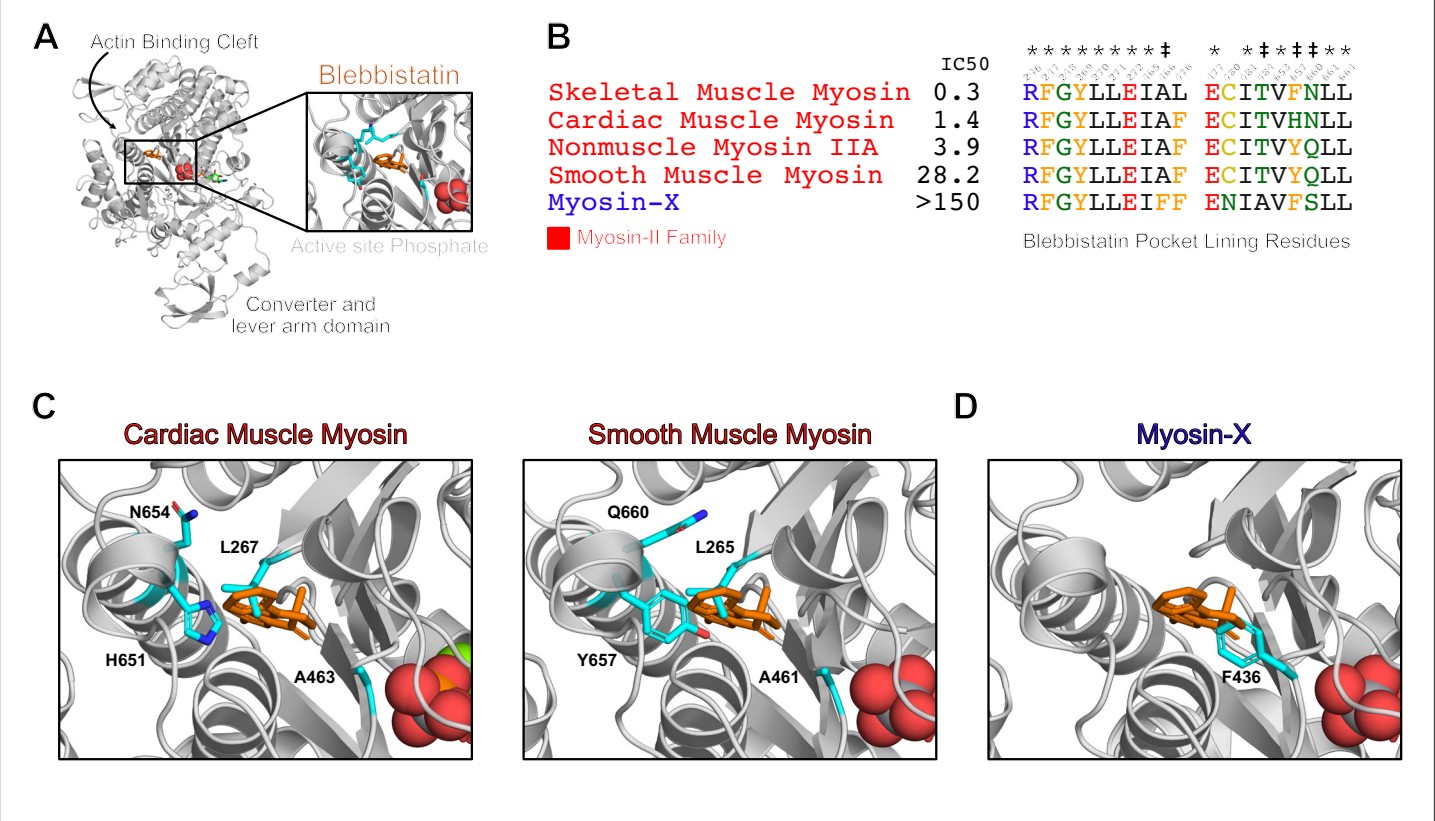

**Figure 1.** Blebbistatin's cryptic binding pocket is closed in blebbistatin-free experimental myosin structures and the sequence of surrounding residues is highly conserved across myosin isoforms with widely varying IC50s. (**A**) The motor domain of a *D.d.* myosin-II molecule bound to blebbistatin. The inset depicts blebbistatin's cryptic binding pocket (PDB: 1YV3[24]). Blebbistatin is shown in orange sticks while the active site phosphate is shown in spheres. Select residues (same as those shown in C) are shown in cyan sticks. (**B**) Alignment of blebbistatin contact residues (within 5 Å of blebbistatin in 1YV3) reveals that 16 of 19 residues are identical among myosin-IIs despite almost two orders of magnitude difference in blebbistatin IC50. We also include an unconventional myosin-X to highlight an important sequence difference at residue 466 (A vs. F) that separates blebbistatin-sensitive (IC50 <150 μM) and blebbistatin-insensitive isoforms (IC50 >150 μM). A star indicates a residue is conserved among all myosin isoforms shown. A double cross is used to indicate sequence differences previously suggested to play an important role in determining blebbistatin specificity (***Allingham et al., 2005***). (**C**) Crystal structures of β-cardiac (PDB: 5N6A ***Planelles-Herrero et al., 2017***) and smooth muscle myosin (1BR2 ***Dominguez et al., 1998***) do not suggest an obvious mechanism for differences in blebbistatin potency between these isoforms. In both structures, the cryptic blebbistatin-binding pocket is closed, and an aligned blebbistatin molecule (orange) clashes with a leucine residue (cyan). The two residues that differ between these isoforms (cyan histidine and asparagine in cardiac; cyan tyrosine and glutamine in smooth) do not form specific interactions with blebbistatin (e.g. hydrogen bonds). (**D**) In a blebbistatin-free myosin-X structure (PDB: 5I0H ***Ropars et al., 2016***), F436 (cyan) forms a large steric impediment to blebbistatin binding (aligned molecule shown in orange) that is not present in myosin-II isoforms.

It has previously been suggested that blebbistatin binds at a 'cryptic' pocket that is usually closed in crystal structures of myosins without a bound blebbistatin. (***Cimermancic et al., 2016***) Indeed, the blebbistatin binding site is closed in crystal structures of two blebbistatin-sensitive myosin isoforms (***Figure 1C***). These results suggest that blebbistatin sensitivity is also encoded by factors beyond the sequence of the binding pocket.

We hypothesized that blebbistatin potency among myosin-II family members is encoded in the ensemble of structures that myosins adopt in solution, with more sensitive isoforms (i.e. lower IC50s) having a higher probability of adopting conformations with an open blebbistatin-binding pocket. A growing body of work has strengthened the view that cryptic pockets closed in crystal structures can open in excited structural states explored in solution (***Zimmerman et al., 2021***; ***Kuzmanic et al., 2020***; ***Bowman et al., 2015***; ***Porter et al., 2019a***; ***Hollingsworth et al., 2019***). Hence, we reasoned that the blebbistatin cryptic pocket would open in all-atom molecular dynamics simulations. To test our hypothesis, we leverage all-atom molecular dynamics simulations, Markov State Models (MSMs),

and a novel MSM-based approach to aggregating docking results across structural ensembles for accurate prediction of binding affinities.

## Results

### Blebbistatin's cryptic binding pocket opens in simulations

We first sought to establish whether the blebbistatin pocket is open in any blebbistatin-free myosin experimental structures or if it is a 'cryptic' site. Cryptic pockets are cavities that open and close as a protein fluctuates in solution but are typically closed and therefore hidden in experimental structures.

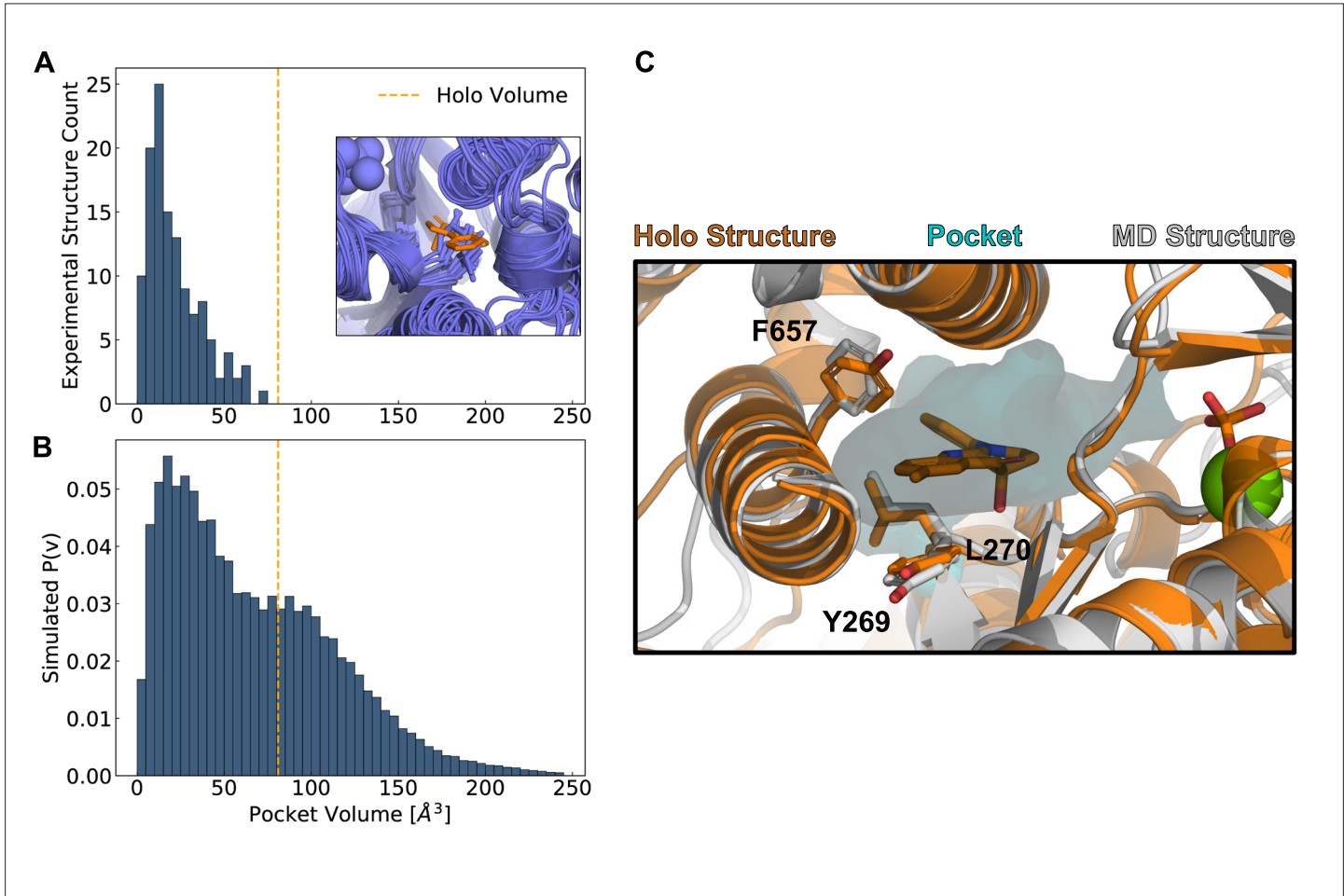

**Figure 2.** Simulations reveal opening of blebbistatin's cryptic pocket. (**A**) The distribution of pocket volumes from experimental crystal structures queried from the Protein Data Bank shows that the blebbistatin pocket is cryptic. The inset is a random selection of 15 structures from the accompanying distribution with an overlaid blebbistatin molecule in orange. All experimentally determined myosin structures display steric clash with a blebbistatin molecule aligned based on its contact residues in a blebbistatin-bound, or *holo*, structure (PDB: 1YV3). (**B**) Blebbistatin pocket volumes in simulations of fast skeletal myosin IIA reveal substantial pocket opening. The blebbistatin pocket volume from a ligand-bound crystal structure (PDB: 1YV3) is delineated by an orange vertical line in both panels. Simulated P(v) corresponds to the probability of adopting a given volume for each bin in the histogram. (**C**) MD simulations explore open *holo*-like states. Structure of an open conformation of the blebbistatin binding pocket from MD simulations reveals good structural alignment with the *holo* crystal structure (0.55 Å root mean square deviation of contact residue backbone heavy atom and Cβ positions). Blebbistatin is shown in orange with the pocket from the MD structure shown as a cyan contour. Selected residues in the blebbistatin pocket (Y269, L270, and F657) have the same backbone and sidechain positions as in the *holo* crystal structure. Note that reported pocket volumes are smaller than the space available to ligands because of an algorithm choice made to avoid erroneous detection of small pockets (see Materials and methods for details).

The online version of this article includes the following figure supplement(s) for figure 2:

**Figure supplement 1.** Trajectory traces for long simulations of skeletal muscle myosin reveal opening in all long MD trajectories.

To assess whether the blebbistatin-binding site is cryptic, we queried the Protein Data Bank (PDB) for an exhaustive set of myosin motor domains, yielding 124 structures that were not crystallized with blebbistatin or blebbistatin derivatives (see Materials and methods). We then assessed the degree of pocket opening at the blebbistatin binding site using the LIGSITE pocket detection algorithm (*Hendlich et al., 1997*). Briefly, LIGSITE finds concavities on a protein surface by identifying grid points that are surrounded by protein but not in contact with protein atoms (see Materials and methods). Those LIGSITE grid points that were within 5 Å of an aligned blebbistatin molecule were considered part of the blebbistatin pocket. The pocket was considered open if its volume matched or exceeded that of the blebbistatin-bound, or *holo*, structure (PDB: 1YV3).

We find that all known blebbistatin-free experimental structures of the myosin motor domain have a closed blebbistatin pocket (*Figure 2A*). Not a single blebbistatin-free experimental structure reaches the *holo* pocket volume, and most blebbistatin-free structures have less than half of the *holo* pocket volume. In blebbistatin-free myosin experimental structures, a leucine residue in the U50 linker (*Allingham et al., 2005*), a highly conserved loop in the upper 50 kDa domain, always points into the blebbistatin pocket, creating a steric impediment to binding (*Figure 2A* inset). While the blebbistatin binding site has previously been annotated as a cryptic pocket, previous analyses were restricted to a subset of blebbistatin-free myosin experimental structures that matched the *holo* structure's sequence exactly (*Cimermancic et al., 2016*). Here, we have shown that all available blebbistatin-free experimental structures of the myosin family lack an open blebbistatin pocket.

Even if the blebbistatin pocket is closed in blebbistatin-free experimental structures, we reasoned that the pocket might open in excited states accessible in all-atom molecular dynamics (MD) simulations. Recent work demonstrates that in solution myosins sample a broad range of conformations driven by thermal fluctuations (*Porter et al., 2020*; *Muretta et al., 2015*), even though all myosin motor domains share a common fold (*Robert-Paganin et al., 2020*). To assess whether the blebbistatin pocket opens in solution without blebbistatin present, we used molecular dynamics to simulate the motor domain of human fast skeletal myosin IIA (*MYH2*). Fast skeletal myosin is potently inhibited by blebbistatin (average IC50: 0.3 μM *Limouze et al., 2004*; *Várkuti et al., 2016*; *Radnai, 2021*), so we hypothesized that its blebbistatin pocket would open extensively in simulation.

We constructed Markov State Models (MSMs) from over 80 microseconds (Table 3) of simulations of the actin-free, ADP-phosphate-bound fast skeletal myosin motor domain. MSMs of molecular simulations are network models of free energy landscapes composed of many conformational states and the probabilities of transitioning between these states (*Bowman et al., 2014*). We constructed MSMs of the conformations seen in the blebbistatin pocket by clustering structures in a kinetically relevant projection of backbone and sidechain dihedral angles (see Materials and methods). To measure blebbistatin pocket opening, we measured pocket volumes at the blebbistatin binding site using the LIGSITE algorithm as described above. We then quantified the probability of pocket opening based on the probability of each structure in the MSM (see Materials and methods).

In contrast to myosin crystal structures, we find that simulations reveal extensive opening of the blebbistatin pocket. The distribution of pocket volumes from simulation is substantially right-shifted relative to the distribution seen in crystal structures (*Figure 2B*). Even though our simulations were started from a closed blebbistatin-free conformation, all 8 long (>500 nanosecond) independent simulations of fast skeletal myosin IIA exceed the volume seen in the blebbistatin-bound crystal structure (*Figure 2—figure supplement 1*). Indeed, the blebbistatin pocket is open in about one-third of conformations at equilibrium ($p_{open}$ = 0.31). When we visually inspected structures from simulation that had reached the *holo* pocket volume (*Figure 2C*), we find that the blebbistatin pocket geometry closely matches that of the blebbistatin-bound crystal structure (the R.M.S.D. for the structure depicted in *Figure 2C* was 0.55 Å when considering the backbone heavy atom and Cβ positions of residues in contact with blebbistatin). While the leucine (L270 in skeletal muscle myosin IIA) in the U50 linker always points into the blebbistatin pocket in blebbistatin-free crystal structures, in simulations this leucine residue rotates toward its blebbistatin-bound position (*Figure 2C*). Thus, we find that simulations can capture blebbistatin cryptic pocket opening that is not seen in myosin crystal structures and that MSMs can be used to quantify the probability of blebbistatin pocket opening.

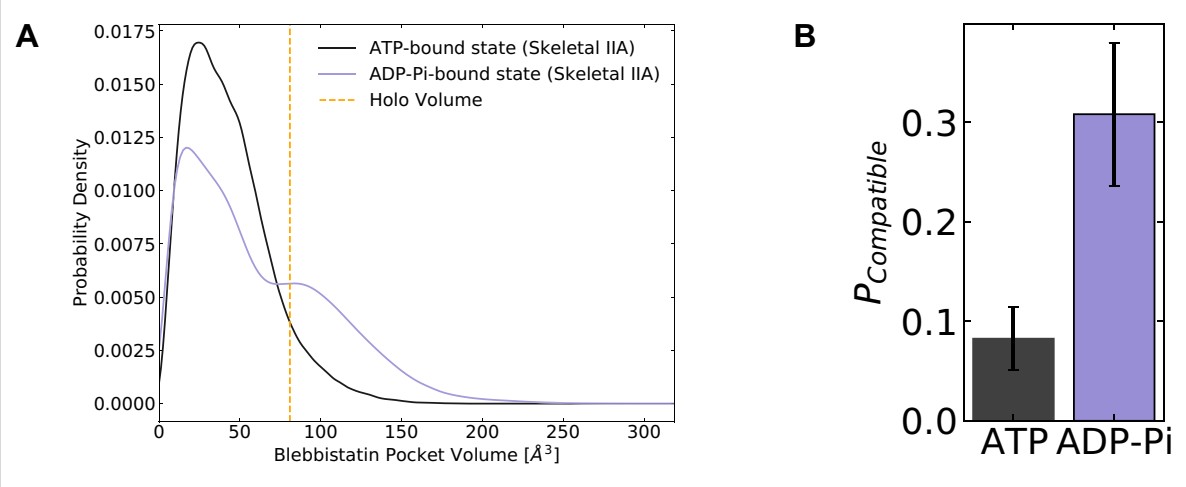

**Figure 3.** Blebbistatin pocket opening preferentially occurs in the ADP*Pi state. (**A**) Distributions of skeletal muscle myosin IIA blebbistatin pocket volumes in the ATP-bound state and ADP*Pi-bound state demonstrate that the blebbistatin pocket is more likely to open in the ADP*Pi-bound state, consistent with biochemical experiments which predict tighter binding between blebbistatin and myosin when myosin is bound to ADP*Pi. (**B**) The probability of adopting compatible structures (i.e. structures with pocket volumes equal to or greater than the blebbistatin-bound crystal structure) is higher when myosin is bound to ADP and phosphate. Error bars represent estimate of standard error of the mean from 250 trials of bootstrapping where trajectories were drawn with replacement from the entire dataset (see Materials and methods). Note that reported pocket volumes are smaller than the space available to ligands because of an algorithm choice made to avoid erroneous detection of small pockets (see Materials and methods for details).

## Blebbistatin's cryptic pocket preferentially opens in the ADP*Pi state

Given that blebbistatin's cryptic pocket opens in simulation, we wondered if pocket opening was dependent on the nucleotide present in the myosin active site. Biochemical experiments have shown that blebbistatin binds to rabbit fast skeletal muscle myosin with ~10-fold greater affinity when myosin is bound to ADP and phosphate instead of ATP (*Kovács et al., 2004*). Hence, we hypothesized that blebbistatin pocket opening would be more likely in simulations of myosin bound to ADP*Pi than it would be in simulation of myosin bound to ATP.

To test this hypothesis, we ran long simulations of human fast skeletal muscle myosin IIA (*MYH2*) from a homology model of a post-rigor crystal structure with ATP in its active site (PDB ID: 6FSA) (*Robert-Paganin et al., 2018*) and assessed opening with LIGSITE as described above. We then quantified the probability of blebbistatin pocket opening in a MSM of the blebbistatin pocket in the ATP-bound state.

We find that while blebbistatin pocket opening occurs in both nucleotide states, it is substantially more probable in simulations of the ADP*Pi state than in the ATP state (*Figure 3A*). In the ADP*Pi state, the equilibrium probability of blebbistatin pocket opening is 0.31. In the ATP state, it is only 0.08 (*Figure 3B*). This finding is consistent with the experimental results showing that blebbistatin is more likely to bind to a myosin bound to ADP and phosphate, trapping it in this state of the mechanochemical cycle (*Ramamurthy et al., 2004*). Furthermore, this dependence between the nucleotide state in the active site and blebbistatin pocket opening indicates that simulations are capturing the allosteric coupling between these two parts of the myosin molecule.

## The probability of cryptic pocket opening predicts trends in blebbistatin potency

We reasoned that an important determinant of how potently blebbistatin inhibits a myosin isoform is how likely the blebbistatin pocket opens. If pocket opening is more likely in one isoform, then stabilizing the open state should be easier than in an isoform where pocket opening is less probable. Thus, we hypothesized that the blebbistatin pocket would be more likely to open in myosin isoforms more potently inhibited by blebbistatin (i.e. those with lower IC50s).

We first assessed whether pocket opening probabilities could distinguish blebbistatin-sensitive isoforms (IC50 <100 μM) from blebbistatin-insensitive isoforms (IC50 beyond detectable limit). To

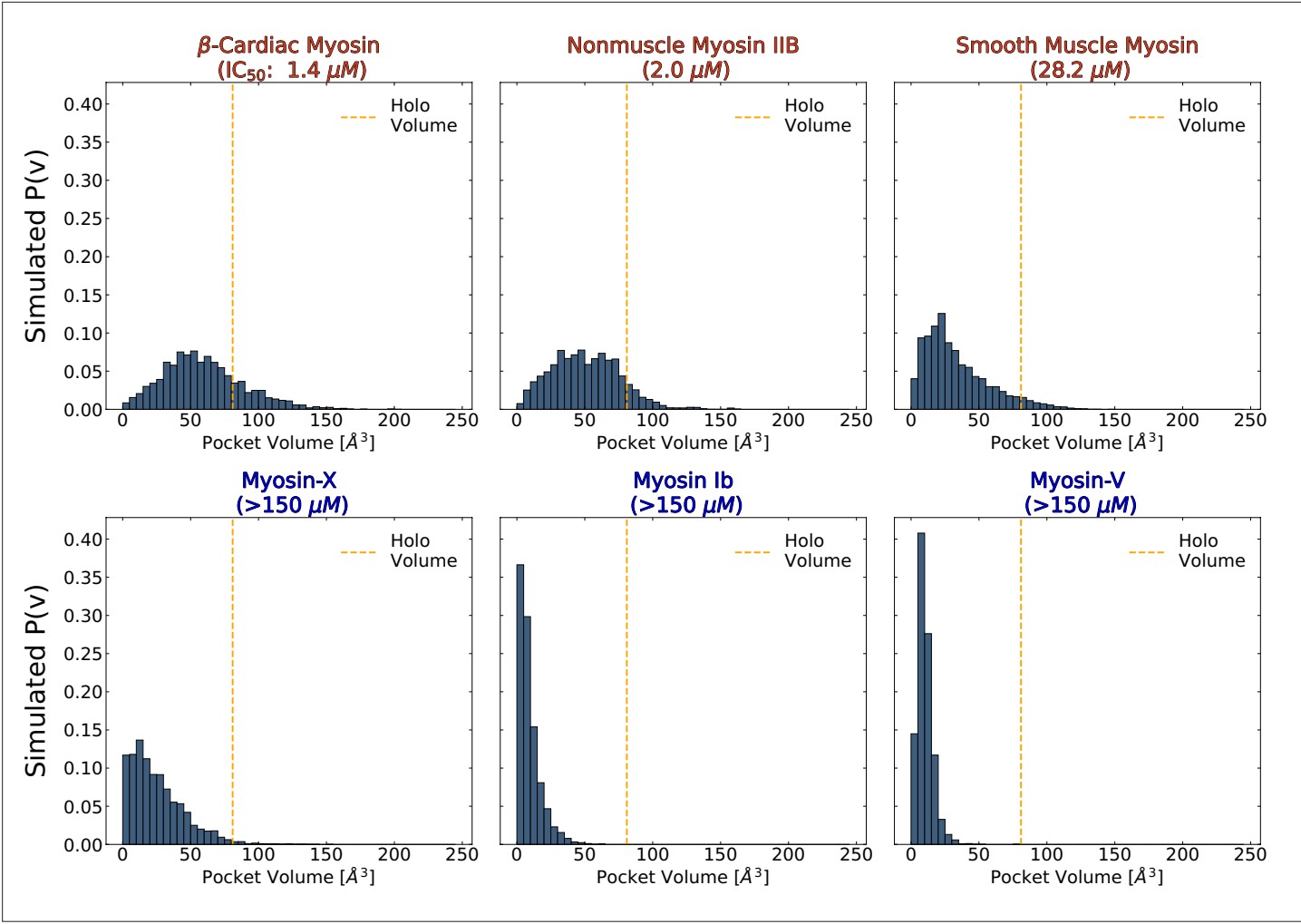

**Figure 4.** The probability of adopting open pocket conformations is greater among blebbistatin-sensitive isoforms (top row) than insensitive isoforms (bottom row). MSM-weighted distributions of blebbistatin pocket volumes in simulations of nucleotide-free isolated myosin motor domains show that myosin-IIs (top row) are more likely to exceed the blebbistatin pocket volume of a *holo* crystal structure (PDB: 1YV3, orange line) than non-class II myosins. Among myosin-IIs, those with lower IC50s (*Limouze et al., 2004*; *Eddinger et al., 2007*; *Wang et al., 2008*; *Zhang et al., 2017*; *Várkuti et al., 2016*; *Radnai, 2021*) have more right-shifted pocket volume distributions. The overall opening probabilities between these isoforms can be visualized in *Figure 4—figure supplement 1*. Note that reported pocket volumes are smaller than the space available to ligands because of an algorithm choice made to avoid erroneous detection of small pockets (see Materials and methods for details).

The online version of this article includes the following figure supplement(s) for figure 4:

**Figure supplement 1.** The probability of adopting open pocket conformations is greater among blebbistatin-sensitive isoforms (red bars) than insensitive isoforms (blue bars).

compare sensitive and insensitive isoforms, we calculated pocket volumes at the blebbistatin binding site in an existing simulation dataset of blebbistatin-free and nucleotide-free myosin motor equilibrium fluctuations (see Materials and methods) (*Porter et al., 2020*). While blebbistatin binds with reduced affinity to nucleotide-free myosins like the ones in these simulations (*Kovács et al., 2004*), our previous work demonstrated that simulations can capture pocket opening in excited states even in the absence of relevant binding partners. We utilized the LIGSITE pocket detection algorithm to assign a blebbistatin pocket volume to each state in the myosin motor MSMs following the same procedure as described above. We then defined compatible states as those conformations where the blebbistatin pocket volume reached or exceeded its volume in a blebbistatin-bound crystal structure of *D.d.* myosin-II (PDB: 1YV3) (*Allingham et al., 2005*).

We find that the probability of adopting a pocket conformation compatible with blebbistatin binding is higher among myosin-II isoforms compared to other isoforms. We observe large differences

in the blebbistatin pocket volume distributions between MSMs of myosin-IIs and non-myosin-IIs (*Figure 4*). In simulations of unconventional myosin-V and myosin-Ib, the blebbistatin pocket stays entirely closed despite almost 300 microseconds of aggregate simulation time per isoform. In contrast, all myosin-IIs sample conformations with pocket volumes that exceed the volume of a blebbistatin-bound crystal structure. Interestingly, among myosin-IIs the probability of pocket opening (smooth muscle myosin <nonmuscle myosin IIb < β-cardiac myosin) correctly predicts the rank order of IC50 values (*Figure 4—figure supplement 1*).

Thus, blebbistatin pocket opening differs between divergent myosin isoforms when there are large sequence differences at the binding site. While myosin-IIs contain small sidechains at the A466 position (skeletal muscle myosin numbering), other myosin families contain aromatic sidechains which point into the pocket, reducing the pocket volume available for blebbistatin binding (*Figure 1B and D*). Moreover, while myosin-II isoforms have a conserved pocket that opens during simulations of nucleotide-free motor domains, the probability of opening appears to correlate with blebbistatin's potency.

To relate the probability of pocket opening more precisely to blebbistatin potency, we ran simulations of several sensitive myosin-II isoforms (β-cardiac myosin, nonmuscle myosin IIA, and smooth muscle myosin) that exhibit a broad range of blebbistatin IC50 values (*Figure 1B*). Since blebbistatin preferentially inhibits myosin when the motor domain is bound to ADP and phosphate (*Kovács et al., 2004*), we ran these MD simulations with ADP*Pi in the active site. Each of the myosin-II isoforms was launched from a closed, blebbistatin-free starting conformation (*Figure 5A*). We observed pocket opening in all simulations of myosin-II isoforms, but the likelihood of opening was substantially enhanced in simulations of fast skeletal muscle myosin IIA and β-cardiac myosin. Indeed, in simulations of both skeletal muscle myosin IIA and β-cardiac myosin, a rolling average (window of 10 ns) of pocket volumes exceeds the *holo* volume for over 500 ns while in simulations of nonmuscle myosin IIA and smooth muscle myosin the pocket only opens transiently a handful of times (*Figure 5*, *Figure 5—figure supplement 2*).

We find that the probability of pocket opening is larger for myosin-II isoforms more potently inhibited by blebbistatin. To quantify probabilities of pocket opening, we constructed a MSM of the conformations seen in the blebbistatin pocket for each isoform separately. We then computed the blebbistatin pocket volume for all structures visited by the simulations, assigned a probability to each structure based on the MSM, and found the overall probability of reaching a pocket volume matching or exceeding that of a *holo* crystal structure (see Materials and methods). Among myosin-IIs, the probability of pocket opening is substantially higher for fast skeletal myosin IIA and β-cardiac myosin than it is for nonmuscle myosin IIA and smooth muscle myosin (*Figure 5E*). Smooth muscle myosin has the lowest probability of pocket opening (~0.0005), so its free energy difference between open and closed states is the largest. Similarly, nonmuscle myosin IIA has a slightly larger but still low probability of pocket opening (~0.001), consistent with its intermediate IC50. Conversely, both fast skeletal myosin IIA ($p_{open}$ = 0.31) and β-cardiac myosin ($p_{open}$ = 0.46) have large probabilities of adopting open states in their MSMs. Given that β-cardiac myosin has a higher opening probability but is less sensitive to blebbistatin, we wondered if volume is a useful, yet incomplete descriptor, for assessing differences in blebbistatin affinity. To test this hypothesis, we turned to molecular docking to see if we could quantitatively predict the binding affinity for blebbistatin between these myosin-II isoforms by using ensembles of structures from our simulations.

## MSM-docking quantitatively predicts blebbistatin's potency

We reasoned that molecular docking could improve our ability to predict blebbistatin's potency by considering the chemical environment of the pocket rather than just the volume available to it. Some of the states we had labeled as closed based on pocket volume, especially those with volumes slightly less than the *holo* crystal structure, might be able to accommodate blebbistatin in alternate poses. In contrast, some open states may be less compatible with binding than others. We have previously shown that docking compounds to a diverse set of conformations from a Markov State Model improves agreement with experiment (*Hart et al., 2016*). In this work, we wished to dock to open and closed structures from the ensemble to generate a robust estimate of the free energy of blebbistatin binding.

To determine whether crystal structures were sufficient to explain differences in blebbistatin potency, we first docked blebbistatin to single open and closed structures found in the PDB. When

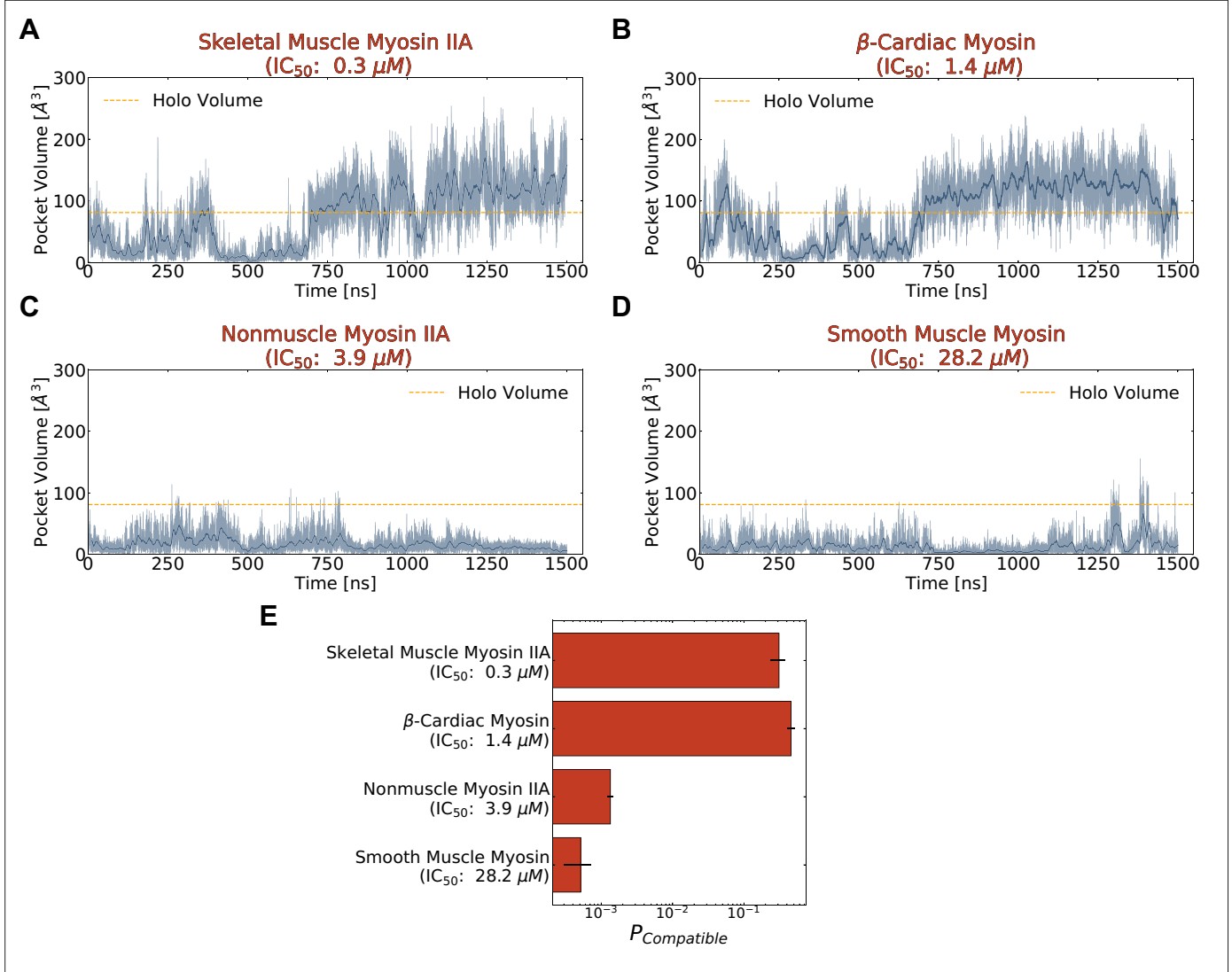

**Figure 5.** Blebbistatin cryptic pocket opening is more likely in simulations of highly sensitive myosin-II isoforms than in simulations of less sensitive myosin-IIs. (**A-D**) Representative pocket volume trajectory traces for several myosin-II isoforms show that pocket opening occurs with greater frequency and that the pocket stays open for longer in those isoforms more potently inhibited by blebbistatin (top row). The dotted orange line delineates the blebbistatin pocket volume in a *holo* crystal structure (PDB ID: 1YV3). Transparent blue lines indicate raw data while the opaque blue lines are a 10 ns rolling average. (**E**) Blebbistatin pocket opening is highly probable (>0.25) in skeletal muscle myosin II and β-cardiac myosin but highly unlikely (<0.01) for nonmuscle myosin IIA and smooth muscle myosin. A conformation was considered compatible if its blebbistatin pocket volume exceeded the pocket volume of a *holo* crystal structure (PDB ID: 1YV3). Conformations were weighted by their equilibrium probability in Markov State Models of the blebbistatin pocket. Error bars show bootstrapped estimate of standard error of the mean from 250 trials. Note that reported pocket volumes are smaller than the space available to ligands because of an algorithm choice made to avoid erroneous detection of small pockets (see Materials and methods for details).

The online version of this article includes the following figure supplement(s) for figure 5:

**Figure supplement 1.** The blebbistatin pocket stays open for prolonged periods (>200 ns) of simulation time in both skeletal and β-cardiac myosin.

**Figure supplement 2.** VAMP-2 (Variational approach for Markov processes) scores for MSMs constructed with varying numbers of cluster centers were computed on a validation set of trajectories to select an appropriate number of cluster centers for each MSM.

**Figure supplement 3.** implied timescales for Markov State Models of the blebbistatin pocket across multiple myosin isoforms show convergence on a logarithmic scale.

we docked blebbistatin to the closed experimental structures from which simulations were launched, we found that it occupied an adjacent cavity but that these poses received very poor docking scores for all myosin-II isoforms (*Figure 6—figure supplement 1*). There were no subtle structural differences between experimental structures or adjacent pockets that might explain variation in blebbistatin potency. We also docked blebbistatin to homology models of the 4 different myosin-IIs considered above in the *holo* crystal structure. This allowed us to interrogate if a single binding-competent structure with the appropriate pocket residues could explain differences in blebbistatin affinity. We found virtually no differences in predicted binding affinities between myosin-IIs using docking to these static structures (*Figure 6—figure supplement 1*).

Next, we docked blebbistatin to the ensemble of structures represented in our MSMs. Specifically, we used AutoDock Vina to dock blebbistatin against representative structures from each state of our MSM within a box centered on the blebbistatin binding site (see Materials and methods). After completing docking, we investigated both the highest scoring poses and those poses with the lowest blebbistatin RMSD from *holo*. Encouragingly, the highest scoring pose for skeletal muscle myosin and β-cardiac myosin is very similar to blebbistatin's pose in the previously determined experimental *holo* crystal structure (RMSD <3 Å for the ligand heavy atoms, *Figure 6—figure supplement 3*). All four myosin-II isoforms had docked poses with ligand heavy atom RMSD <3 Å from the *holo* pose, but the conformational ensembles of skeletal muscle and β-cardiac myosin have a substantially higher probability of adopting conformations where blebbistatin docks in *holo*-like poses (*Figure 6—figure supplement 2*).

We find that computationally predicted blebbistatin binding free energies based on state populations from our Markov State Models closely match experimental values. To calculate a blebbistatin binding free energy for each isoform, we assigned a probability to each structure from docking based on the overall probability of that structure's MSM state and the number of other structures that were mapped to that MSM state (see Materials and methods). Finally, we aggregated the docking results by finding a weighted average of binding constants and converting this value to a free energy of binding. To assess the accuracy of these predictions, we pooled IC50 measurements from all available experiments and converted these measurements to binding free energies under the assumption that IC50 was essentially equivalent to $K_i$ (see Materials and methods for detailed rationale). When we compared the predicted binding affinity of blebbistatin from docking to experimental averages, we find that these parameters are well correlated (*Figure 6A*, $R^2$=0.82). Moreover, we note that the absolute value of our predictions of binding free energy are in good agreement with binding affinities estimated from experiment (root mean square error of 0.7 kcal/mol). Thus, our results suggest that docking to the ensemble of structures with MSM weighting provides an accurate way to rapidly assess the relative binding affinity of blebbistatin across myosin isoforms.

To test our ability to make blind predictions with this approach, we computationally and experimentally interrogated blebbistatin inhibition of a sarcomeric myosin-II isoform called Myh7b whose sensitivity to blebbistatin had not been determined. Myh7b's blebbistatin binding site is identical to β-cardiac myosin's. However, there are numerous sequence differences immediately surrounding blebbistatin's binding site (7 positions that differ between Myh7b and β-cardiac myosin within 1 nm of blebbistatin's binding site). In simulations of a homology model of its motor domain, Myh7b had substantial pocket opening (*Figure 6—figure supplement 5*). Moreover, when we docked to representative structures from the Myh7b simulations and aggregated predicted binding affinities using its MSM, we predict that the binding affinity of blebbistatin for Myh7b is –8.8 kcal/mol. Thus, we hypothesized that Myh7b would be highly sensitive to blebbistatin inhibition and that its IC50 would be more similar to the IC50 of fast skeletal and β-cardiac myosin than that of smooth muscle myosin.

We find that blebbistatin potently inhibits the actin-activated ATPase activity of Myh7b (*Figure 6B*, IC50: 0.36 μM). We used recombinant human Myh7b and β-cardiac myosin S1 constructs expressed in $C_2C_{12}$ cells in our experiments. We measured NADH-linked ATPase rates at increasing concentrations of blebbistatin (0.3125 μM to 20 μM) and fit a hyperbolic Hill equation to the data to determine the IC50. As a control, we measured the IC50 value for β-cardiac myosin S1. We obtain an IC50 of 1.12±0.29 μM (error indicates standard deviation between replicates), a value which closely matches with previously published IC50s (*Figure 6—figure supplement 6*). Consistent with our hypothesis and the computational prediction of blebbistatin's affinity for Myh7b based on MSM-docking (0.67 μM), we obtain an experimentally measured IC50 for Myh7b (0.36±0.08 μM) that is more similar to fast skeletal

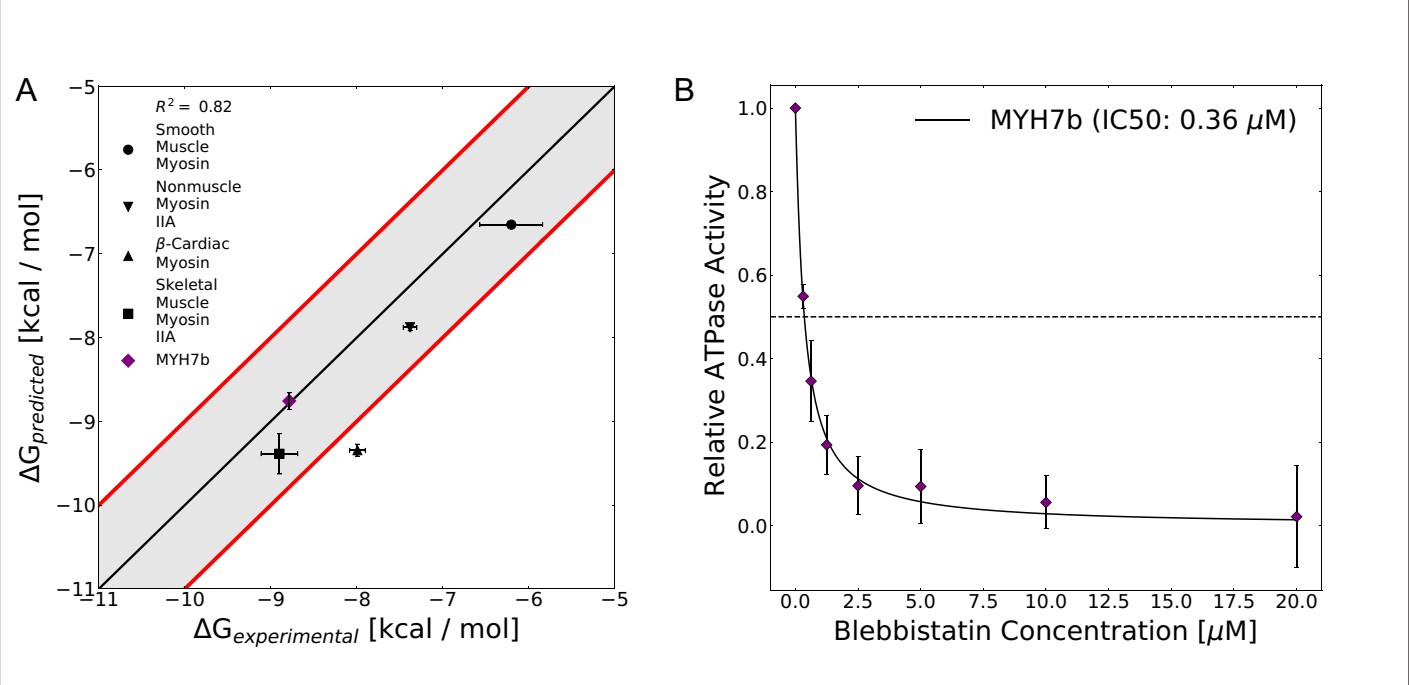

**Figure 6.** The computed free energy of binding for blebbistatin from MSM-docking accurately predicts binding free energies for existing experimental data and for a myosin isoform whose blebbistatin sensitivity was not known. (**A**) Predictions from MSM-docking are highly correlated to experimental values ($R^2$=0.82) and most predictions are within 1 kcal/mol of experimental values. Error bars for predicted free energies of binding represent bootstrapped estimate of standard error of the mean from 250 trials. Error bars for experimental values show the standard error of the IC50 or $K_i$ converted to a binding free energy. (**B**) An NADH-linked ATPase assay indicates that MYH7b is highly sensitive to blebbistatin inhibition (IC50: 0.36 μM), consistent with the prediction from MSM-docking. Data show the mean ATPase activity ± standard deviation across 5 experimental replicates (2 biological replicates, each with two or three technical replicates).

The online version of this article includes the following figure supplement(s) for figure 6:

**Figure supplement 1.** Docking scores to homology models of *apo* and *holo* structures do not correlate with blebbistatin potency.

**Figure supplement 2.** Comparison of distribution of docking scores and ligand heavy atom root mean square deviation (RMSD) from blebbistatin's pose in a *holo* crystal structure (PDB: 1YV3) show that skeletal and β-cardiac myosin are more likely to adopt structures where blebbistatin can be docked in its *holo* orientation and obtain a favorable docking score.

**Figure supplement 3.** Highest scoring poses for each of the myosin isoforms reveals that the best pose for skeletal muscle myosin, β-cardiac myosin, and Myh7b closely matches the pose seen in *holo* crystal structures (ligand heavy atom RMSD 1.2 Å, 1.5 Å, and 0.9 Å to *holo* PDB 1YV3 for skeletal muscle myosin, β-cardiac myosin, and Myh7b, respectively).

**Figure supplement 4.** Docking to nonmuscle myosin IIA and smooth muscle myosin produces low RMSD poses (3.2 Å and 3.0 Å blebbistatin heavy atom RMSD from holo 1YV3 structure) with reasonably high scores (–6.3 kcal/mol in both cases).

**Figure supplement 5.** MSM-weighted pocket volumes for myosin-II isoforms in the ADP*Pi state reveal that blebbistatin pocket opening commonly occurs in skeletal muscle myosin, β-cardiac myosin, and Myh7b.

**Figure supplement 6.** Blebbistatin inhibits the actin-activated ATPase activity of β-cardiac myosin with an IC50 of 1.12 μM.

and β-cardiac myosin's IC50 than smooth muscle myosin's IC50. Thus, our experiments provide additional validation for the MSM-docking approach.

## Discussion

A ligand's specificity is most typically attributed to differences in the composition of residues at the ligand binding site. Sequence variation at a binding site can modify the shape of a binding pocket or alter the types of interatomic interactions formed between a ligand and its target protein. Mutagenesis experiments have shown that sequence differences in the blebbistatin pocket are an important determinant of blebbistatin's selectivity for myosin-IIs (**Zhang et al., 2017**). While myosin isoforms outside the myosin-II family have a bulky aromatic residue that points into the blebbistatin pocket,

myosin-IIs have smaller residues at that analogous position (*Figure 1D*). However, the binding pocket residues that coordinate binding of blebbistatin are insufficient to fully explain its isoform specificity.

Our results highlight that the distribution of structural states explored in solution can be an essential determinant of specificity. We find that differences in blebbistatin pocket dynamics are important determinants of differences in IC50 between myosin-II isoforms. Pocket opening is substantially more probable in skeletal muscle myosin than it is in smooth muscle myosin, consistent with blebbistatin's more potent effects on skeletal muscle myosin. Furthermore, even when the blebbistatin binding site is perfectly conserved between isoforms, such as in the cases of nonmuscle myosin-IIA and smooth muscle myosin, MSM-weighted docking predicts differences in blebbistatin affinity which are consistent with experimentally measured differences in IC50. Thus, pocket dynamics, together with differences in the blebbistatin pocket's residue composition are both important determinants of specificity.

Given there are differences in pocket opening probabilities even when pocket-lining residues are identical or near identical, distant sequence elements must allosterically modulate pocket opening. Our results demonstrate that physics-based simulations can capture these allosteric networks by modeling complex couplings throughout the myosin molecule. Even though myosin motors from the myosin-II family share high-sequence identity (*Supplementary file 1*) and structural similarity, they are evolutionarily tuned to explore different distributions of conformations in solution (*Porter et al., 2020*). Future work is needed to isolate precisely which sequence elements control blebbistatin pocket opening.

Though our present work relies on combining long simulations and highly parallel simulations gathered on folding@home with docking, we speculate that several alternative methods could produce similar results. Efficient sampling of cryptic pocket opening is possible with adaptive sampling methods like FAST (*Zimmerman and Bowman, 2015*) or enhanced sampling methods (*Henin et al., 2022*) like metadynamics (*Meller et al., 2022*) and accelerated molecular dynamics (*Miao et al., 2016*). In many cases, a relatively modest amount of sampling is sufficient to observe cryptic pocket opening (*Meller et al., 2023*)*Meller et al., 2022*, although accurate estimation of opening probabilities may require additional sampling. Furthermore, we note that our analysis could be improved by using more rigorous, physics-based methods for predicting protein-ligand affinity (e.g. Molecular Mechanics Generalized Boltzmann Surface Area *Wang et al., 2019*). Future work will investigate which sampling strategies and binding prediction methods are most efficient and accurate.

Our findings agree with other studies that demonstrate the importance of pocket dynamics in modulating ligand specificity. Another myosin inhibitor, CK-571, which specifically targets smooth muscle myosin binds at a perfectly conserved binding site, suggesting an important role for structural dynamics (*Sirigu et al., 2016*). Furthermore, the CK-571 pocket has not been observed in ligand-free myosin structures. Similar results have been found in other systems. Positive allosteric modulators of a G-protein-coupled receptor bind at a dynamic cryptic pocket (*Hollingsworth et al., 2019*), and their selectivity has been attributed to differences in cryptic pocket opening. Together, a growing body of evidence suggests that cryptic pockets can be exploited to develop isoform-specific drugs against proteins with nearly identical crystal structures.

Additionally, our finding that the blebbistatin pocket readily opens in simulation is consistent with an important role for conformational selection. Many other simulation studies have also demonstrated that protein conformational changes associated with ligand binding are present in the ligand-free ensemble of structures adopted by proteins in solution (*Meller et al., 2023*; *Bowman and Geissler, 2012*; *Oleinikovas et al., 2016*). Furthermore, a growing body of experimental evidence supports a primary role for conformational selection (*Changeux and Edelstein, 2011*; *Pelc et al., 2022*; *Vogt and Di Cera, 2012*). However, there are examples (e.g. Niemann-Pick Protein C2) when even several rounds of adaptive sampling simulations do not sample cryptic pocket opening. These results along with a flux-based analysis of binding mechanisms suggest that induced fit may also play an important role, especially at high ligand concentrations (*Hammes et al., 2009*).

Finally, our results highlight the general capacity of computational modeling to capture how subtle sequence differences induce conformational preferences, which, in turn, can control function. Simulations that reveal how sequence variation impacts conformational dynamics have potential to bolster our understanding of how patient-specific mutations contribute to protein dysfunction and drug response as well as to guide the development of new therapeutics. Indeed, certain myosin variants associated with disease may show mutation-induced changes in dynamics that could be targeted as

**Table 1.** Homology models prepared for this study.

| Isoform | Gene | UniprotID | Template structure | Structural state / Nucleotide state |
|---|---|---|---|---|
| Fast Skeletal | MYH2 | Q9UKX2 | 5N6A | PPS (ADP*Pi) |
| β-Cardiac | MYH7 | P12883 | 5N6A | PPS (ADP*Pi) |
| Nonmuscle IIA | MYH9 | P35579 | 5I4E | PPS (ADP*Pi) |
| Smooth | MYH11 | P35749 | 1BR2 | PPS (ADP*Pi) |
| MYH7b | MYH7B | A7E2Y1 | 5N6A | PPS (ADP*Pi) |
| Fast Skeletal | MYH2 | P12883 | 6FSA | PR (ATP) |

PPS indicates prepowerstroke while PR indicates post rigor.

part of a precision medicine approach (*Greenberg and Tardiff, 2021*; *Snoberger et al., 2021*). Thus, our work represents an important advancement toward physics-based precision medicine.

# Materials and methods

## Key resources table

| Reagent type (species) or resource | Designation | Source or reference | Identifiers | Additional information |
|---|---|---|---|---|
| Gene (*H. sapiens*) | β-MyHC S1 | UNIPROT: P12883 | | amino acids 1–842 |
| Gene (*H. sapiens*) | MYH7b S1 | PMID:36334627 | | amino acids 1–850 |
| Cell line (*M. musculus*) | C2C12 cells | PMID:20080549 ATCC | | |
| Chemical compound, drug | Blebbistatin | Selleckchem | S7099 | |
| Software, algorithm | GROMACS | https://doi.org/10.1016/j.softx.2015.06.001 | | Version 2021.1 |

## Structural bioinformatics

We queried the PDB for all experimental structures of myosin motor domains with a sequence identity cutoff of 10% to *hs MYH7*, resolution ≤ 4.0 Å, and a BLAST E-value less than 0.1. All PDBs satisfying these criteria were downloaded for further analysis. Some of these PDB files contained fragments of motor domains; therefore, the resulting database of PDBs was parsed further by selecting the largest chain in each crystal structure if the sequence was >600 amino acids. Structures containing blebbistatin, or blebbistatin derivatives, were also excluded from the set (PDB: 6Z7U, 6YSY, 3MJX, 3BZ8,3BZ7, 3BZ9, 1YV3, 3MYK, 3MYH). In total, our set included 124 myosin experimental structures and 23 unique myosin sequences (11 of which were from the myosin-II family).

A multiple sequence alignment was performed with 1YV3 as reference. The resulting alignment was used to identify ligand binding site for the blebbistatin binding pocket and pocket volumes were

**Table 2.** Sequence similarity between sequence used for homology modeling and template structures.

| Gene | Template structure | Sequence identity | Sequence similarity |
|---|---|---|---|
| MYH2 | 5N6A | 80% | 88% |
| MYH7 | 5N6A | 96% | 98% |
| MYH9 | 5I4E | 78% | 90% |
| MYH11 | 1BR2 | 94% | 96% |
| MYH7B | 5N6A | 69% | 85% |
| MYH2 | 6FSA | 80% | 88% |

calculated on each of these structures with the LIGSITE pocket detection algorithm (*Hendlich et al., 1997*) with a minimum rank of 6, probe radius of 0.14 nm, and a minimum cluster size of 3 grid points.

## Preparation of homology models

Initial structural models (i.e. starting structures) for each myosin isoform were generated with homology modeling using SWISS-MODEL (*Waterhouse et al., 2018*). Crystal structures for simulations were selected based on their sequence similarity to the isoform of interest. Higher resolution structures were prioritized. For 5N6A, the converter and N-terminal regions of the protein were replaced with the corresponding converter and N-terminal regions from 5N69 because these regions were poorly resolved in 5N6A. Below, the UniprotID of the respective human myosin isoform are provided as well as the relevant crystal structures used for modeling (*Table 1*).

*Table 2* provides the sequence identity and sequence similarity between the modeled sequence and template structure's sequence.

We note that the prepowerstroke state structures of myosin-IIs share very high structural similarity (less than 1 Å in C-α RMSD, *Supplementary file 1*).

## Molecular dynamics simulations

GROMACS (*Abraham et al., 2015*) was used to prepare and to simulate all proteins using the CHARMM36m force fields (*Huang et al., 2017*). The protein structure was solvated in a dodecahedral box of TIP3P water (*Jorgensen et al., 1983*) that extended 1 nm beyond the protein in every dimension. Thereafter, sodium and chloride ions were added to the system to maintain charge neutrality and 0.1 M NaCl concentration. Each system was minimized using steepest descent until the maximum force on any atom decreased below 1000 kJ/(mol x nm). The system was then equilibrated with all atoms restrained in place at 300 K maintained by the Bussi-Parinello thermostat (*Bussi et al., 2007*) and the Parrinello-Rahman barostat (*Parrinello and Rahman, 1981*).

Production simulations were performed in the CHARMM36m forcefield. Simulations were run in the NPT ensemble at 310 K using the leapfrog integrator, Bussi-Parinello thermostat, and the Parrinello-Rahman barostat. A 12 Å cutoff distance was utilized with a force-based switching function starting at 10 Å. Periodic boundary conditions and the PME method were utilized to calculate the long-range electrostatic interactions with a grid density greater than 1.2 Å$^3$. Hydrogen bonds were constrained with the LINCS algorithm (*Hess et al., 1997*) to enable the use of a constant integration timestep of 2 fs.

Molecular dynamics simulations were initially performed in parallel from single starting structures first on our in-house supercomputing cluster or on Oracle Cloud Infrastructure using a combination of Tesla P100, Quadro RTX 6000, and NVIDIA RTX A5000 nodes. Five starting structures were obtained from RMSD clustering this initial trajectory data based on the pocket backbone and C-β positions. These starting structures were then used for additional simulations on Folding@home (*Shirts and Pande, 2000*) (750 clones initiated with different velocities for each starting structure).

## Markov State Models

To construct a Markov State Model (*Bowman et al., 2014*) of the blebbistatin pocket, we first defined a subset of features that were relevant to blebbistatin pocket opening. We used backbone (phi, psi) and sidechain dihedrals of residues within 5 Å of the blebbistatin molecule as an input set of features describing the blebbistatin pocket.

To perform clustering in a kinetically relevant space, we applied time-structure-independent component analysis (tICA) to these features (*Pérez-Hernández et al., 2013*). Specifically, we used a tICA lag time of 10 ns and retained the top n tICs for each isoform that accounted for 90% of kinetic variance using commute mapping.

To determine the number of microstates in our Markov State Model, we used a cross-validation scheme where trajectories were partitioned into training and testing sets (*McGibbon and Pande, 2015*). Clustering into *k* microstates was performed using only the training set, and the test set trajectories were assigned to these *k* microstates based on the their Euclidean proximity in tICA space to each microstate's centroid. Using the test set only, an MSM was fit using maximum likelihood estimation (MLE) (*Prinz et al., 2011*) and the quality of the MSM was assessed with the rank-10 VAMP-2 score of the transition matrix (*Wu and Noé, 2020*).

**Table 3.** Simulation length and Markov State Model hyperparameters for myosin isoforms.

| System | Structural state | Number of cluster centers | Lag time (ns) | Total simulation time (μs) | Median trajectory length (ns) | Maximum trajectory length (ns) |
|---|---|---|---|---|---|---|
| Skeletal Muscle Myosin | ADP*Pi | 50 | 5 | 89.0 | 21.2 | 1500 |
| β-Cardiac Muscle Myosin | ADP*Pi | 100 | 5 | 90.1 | 21.0 | 1500 |
| Nonmuscle Myosin IIA | ADP*Pi | 100 | 8 | 86.7 | 20.0 | 1500 |
| Smooth Muscle Myosin | ADP*Pi | 50 | 5 | 87.0 | 20.0 | 1500 |
| Skeletal Muscle Myosin | ATP | 100 | 5 | 2.7 | 910.0 | 925 |
| Myosin 7b | ADP*Pi | 100 | 5 | 100.8 | 375.0 | 1500 |

Finally, Markov state models of the blebbistatin pocket were fit for each isoform separately using MLE. Lag times were chosen by the logarithmic convergence of the implied timescales test (*Pande et al., 2010*). MSM construction was performed using the PyEMMA software package (*Scherer et al., 2015*). Details for each model can be found in *Table 3*.

To compare blebbistatin pocket opening in myosin-IIs with opening in other myosin families, we have also used an existing dataset of myosin motor domain MSMs. These MSMs were constructed from AMBER03 force field simulations gathered on folding@home. Clustering was performed using the Euclidean distance between residue sidechain solvent accessible surface area as a distance metric. MSMs were fit for each isoform using row normalization after applying a 1 /n pseudocount. Full simulation, clustering, and MSM construction details can be found in *Porter et al., 2020*.

## Pocket analysis

We calculated pocket volumes for the blebbistatin pocket using the LIGSITE algorithm (*Hendlich et al., 1997*). Specifically, we used the LIGSITE implementation in the enspara software package (*Porter et al., 2019b*) with a minimum rank of 6, probe radius of 0.14 nm, and a minimum cluster size of 3 grid points. We chose a large probe radius to avoid the erroneous detection of pockets that are too small to be relevant for ligand binding. For a grid point to be part of a pocket, LIGSITE requires it be further than the probe radius and van der Waals radius of nearby protein atoms (i.e. its minimum distance to the nearest protein atom must be greater than the sum of the closest atom's van der Waals radius and the probe radius). As a result, our reported volumes represent the core of a pocket and are smaller than the space available to ligands. After generating pocket grid points for a myosin structure, we filtered those grid points in the blebbistatin pocket if they were within 2.5 Å of an aligned blebbistatin molecule. We employed a local alignment using homologous residues within 5 Å of blebbistatin. Finally, we required that pockets be continuous and selected the largest cluster of grid points defined as having a shortest inter-grid point distance of 1.5 Å.

To generate distributions of pocket volumes, we followed two different procedures. For the previously published dataset (*Porter et al., 2020*), we calculated volumes for each representative structure in the Markov State Model and weighted by the equilibrium probability of each state in the MSM. This was done because (a) the size of the dataset prohibited calculating pockets for all simulation frames and (b) these MSMs contained thousands of states and hence were likely to capture a substantial degree of heterogeneity in the blebbistatin pocket. For the new simulations generated for this work, we calculated pocket volumes for every structure we saved from our simulations (save rate of one frame per 20 ps) and then weighted each of these volumes by the probability of a given structure in that isoform's MSM, specifically the equilibrium probability of the MSM state that the structure is assigned to divided by the number of structures assigned to that MSM state. This second approach allows us to track the temporal evolution of pocket volumes in individual trajectories.

## Docking

Docking against individual structures was performed using smina (*Koes et al., 2013*; *Trott and Olson, 2010*). For each MSM state, we randomly extracted either 3 different structures from that state or $\pi_i * 2000$ different structures if that number exceeded 3/2000 (where $\pi_i$ is the equilibrium

probability of the MSM state). PDB files were converted to PDBQT files using AutoDockFR (ADFR suite). The ligand PDBQT files were generated using the same ADFR suite. The ligand charges were assigned using antechamber. To center the docking grid box on the blebbistatin binding pocket, we first selected backbone heavy atoms from residues within 0.5 nm of blebbistatin in its *holo* structure (PDB: 1YV3) and aligned this selection using an iterative procedure described in *Grossfield et al., 2007*. We then used the centroid of the average structure from the final alignment as the center of our box— (0,0,0) in that coordinate system. All alignment and frame selection was done using LOOS (*Romo et al., 2014*; *Romo and Grossfield, 2009*) For the docking search, we set the exhaustiveness to 8 and used the smina scoring function. Jug (*Coelho, 2017*) was used to parallelize docking while gnu parallel (*USENIX, 2022*) was used to parallelize receptor parameterization.

The overall free energy of binding from docking to an MSM can be written as:

$$\Delta G_{total} = -RTln\left(\sum \pi_i K_{eqi}\right)$$

where $\pi_i$ is the equilibrium probability of the MSM state and $K_{eqi}$ is the equilibrium association constant for a MSM microstate calculated from the docking score for that MSM state. Since the scoring function returns docking scores in kcal/mol, it is straightforward to convert to $K_{eqi}$.

We note that this formula directly considers the configurational entropy of the protein by taking a weighted average of the micro-association constants over all the states in the *apo* MSM. Closed states will receive unfavorable (i.e. more positive) docking scores reflecting in small association constants while open states will receive favorable (i.e. more negative) docking scores resulting in large association constants. Similarly, high probability states will contribute more to the weighted sum of micro-association constants, which is the macro-association constant of the ligand binding reaction. Hence, a protein that pays a high entropic penalty for ligand binding because it has only a few, low probability open states will have a small macro-association constant and an unfavorable overall MSM-docking score. Conversely, a protein that pays a low entropic penalty for ligand binding because of its many, high probability open conformers will have a large macro-association constant and hence a more favorable overall MSM-docking score.

Because we docked to multiple structures for each MSM state, we found a consensus docking score by using the following equation:

$$\Delta G_{total} = -k_BTln\left(\sum \frac{\pi_{s_f}}{N} K_{eqi}\right)$$

where f is a structure from simulation, s is the MSM state that structure belongs to, N is the number of structures from that MSM state for which docking was performed, and π is the equilibrium probability of that MSM state.

## Protein expression and purification

Recombinant myosin was produced as previously described (*Sommese et al., 2013*; *Deacon et al., 2012*; *Lee et al., 2023*; *Resnicow et al., 2010*) with minor changes. Adenoviruses encoding human β-MyHC S1 (amino acids 1–842) and human MYH7b S1 (amino acids 1–850) followed by a flexible GSG linker and C-terminal PDZ binding peptide (RGSIDTWV) were used to infect differentiated $C_2C_{12}$ cells. The source of the cell lines was ATCC. They were not authenticated or tested for mycoplasma because they were exclusively used to produce protein, and their identity does not impact any of the results in the manuscript. $C_2C_{12}$ cells were harvested 4 days after infection, flash frozen in liquid nitrogen, and stored at –80 °C. Cell pellets were thawed and lysed using dounce homogenization in 50 mM Tris pH 8.0, 200 mM NaCl, 4 mM $MgCl_2$, 0.5% Tween-20, 5 mM DTT, 1 mM ATP, 0.2 mM PMSF, and 1 X protease inhibitor cocktail (Millipore Sigma/Roche, 11873580001). Lysates were clarified by centrifugation at 39,000 x g for 25 min at 4 °C. The supernatant was filtered through 5 μM and 1.2 μM filters and applied to a column containing SulfoLink resin (ThermoFisher, 20402) coupled to PDZ. The column was washed with 30 mM Tris pH 7.5, 50 mM KCl, 5 mM $MgCl_2$, 1 mM DTT, and 1 mM ATP and myosin S1 (bound by endogenous $C_2C_{12}$ light chains) was eluted using a peptide with tighter affinity for PDZ (WQTWV). Proteins were dialyzed against a storage buffer containing 20 mM MOPS pH 7.0, 25 mM KCl, 5 mM $MgCl_2$ and 10% sucrose, flash frozen in liquid nitrogen, and stored at –80 °C.

Actin was purified from porcine ventricles as previously described (*Greenberg et al., 2014*; *Clippinger et al., 2019*). The concentration of actin was determined spectroscopically as previously described (*Greenberg et al., 2014*; *Clippinger et al., 2019*).

## NADH-linked ATPase measurements

Actin-activated ATPase rates were measured across a range of blebbistatin concentrations using the NADH-coupled assay in a 96-well plate (*De La Cruz and Ostap, 2009*) with a 0.1 µM myosin S1 concentration and 10 µM actin concentration. Before the experiment, actin was polymerized by dialysis in ATPase buffer containing 20 mM Imidazole, 10 mM KCl, 2 mM MgCl$_2$, and 1 mM DTT followed by 1.1 x molar ratio phalloidin stabilization. Experiments were conducted in ATPase buffer with the addition of the NADH-coupled enzymes (0.5 mM phospho(enol)pyruvate [Sigma, P0564], 0.47 mM NADH [Sigma, N7410], 100 U/mL pyruvate kinase [Sigma, P9136], and 20 U/mL lactate dehydrogenase [Sigma, L1254]). Blebbistatin (Selleckchem, S7099) was dissolved in DMSO. The blebbistatin concentration was varied using serial dilutions. Before gathering data, 2 mM ATP was added to each well. Experiments were performed at 25 ° C using a BioTek Syngergy H1 microplate reader. Absorbance was monitored at 340 nm and it decreased linearly with time. Rates for each well were determined based on the linear fitting of the absorbance as a function of time. A control well containing actin, no myosin, and 20 µM blebbistatin was used as a baseline. Finally, a Hill equation was fit to the data to determine an IC50 for each experiment. Each data point consists of five technical replicates.

## Statistical analysis

Bootstrapping was performed to generate error bars for each of the reported simulation measurements. Specifically, we performed 250 trials where we drew N trajectories with replacement from each set of N trajectories, constructed a MSM with the drawn trajectories, and recomputed the observable of interest (e.g. MSM-weighted docking score).

## IC50 to K$_i$ Conversion

The mechanism by which blebbistatin inhibits skeletal muscle myosin's actin-activated ATPase activity has been characterized in detail (*Kovács et al., 2004*). These experiments indicate that blebbistatin binds with the highest affinity to myosin when it is in its ADP*Pi state capable of weakly binding to actin but also has non-negligible affinity for myosin in its ATP-bound state. Thus, blebbistatin can be considered a mixed inhibitor of actin-activated ATPase activity. The K$_m$ for actin activation of S1 ATPase was 24 µM while the actin concentration used to determine blebbistatin's IC50 for skeletal muscle myosin was 43 µM (*Kovács et al., 2004*). Given that blebbistatin's affinity for the ADP*Pi state is 10 x that of its affinity for the ATP-bound state and that $\frac{K_m}{S} = 0.56$, we conclude that IC50 is essentially equal to $K_i$ for skeletal muscle myosin (*Cheng and Prusoff, 1973*). We assume that the same mechanism of inhibition applies to the other myosin-II isoforms and that IC50s for other myosin-II isoforms can be directly converted to $K_i$'s. Thus, we pooled reported IC50s with $K_i$'s across experiments and converted to binding free energies under these assumptions.

## Code and data availability

The code used for the generation, analysis, and visualization of the molecular dynamics data is available via a Github repository at https://github.com/bowman-lab/blebbistatin-specificity; *Meller, 2022*. The dataset corresponding to this repository is at https://osf.io/cv6d2/. To generate *Figure 4*, the existing dataset found here (https://osf.io/54g7p/) was used.

## Acknowledgements

We acknowledge the Folding@home community for its support and generous donation of computing resources. We thank AMD for the donation of critical hardware and support resources from its HPC Fund that enabled many of the computations for this work. We also thank Oracle Cloud Infrastructure for its donation of computational resources. AM was supported by the National Institutes of Health F30 Fellowship (1F30HL162431-01A1). JML was supported by the National Science Foundation (DGE2139839). This work was funded by NSF CAREER Award MCB-1552471 and NIH grants R01 GM124007 and RF1AG067194. G.R.B holds a Packard Fellowship for Science and Engineering from The David & Lucile Packard Foundation. This work was also supported by the National Institutes of

Health (R01 HL141086 to M.J.G. and NIGMS GM 29090 to L.A.L.) and the Children's Discovery Institute of Washington University and St. Louis Children's Hospital (PM-LI-2019–829 M.J.G.).

## Additional information

### Funding

| Funder | Grant reference number | Author |
|---|---|---|
| National Institutes of Health | 1F30HL162431-01A1 | Artur Meller |
| National Institutes of Health | R01 GM124007 | Gregory R Bowman |
| National Institutes of Health | RF1AG067194 | Gregory R Bowman |
| National Institutes of Health | R01 HL141086 | Michael J Greenberg |
| National Institutes of Health | NIGMS GM29090 to L.A.L | Leslie A Leinwand |
| National Science Foundation | DGE2139839 | Jeffrey M Lotthammer |
| National Science Foundation | MCB-1552471 | Gregory R Bowman |

The funders had no role in study design, data collection and interpretation, or the decision to submit the work for publication.

### Author contributions

Artur Meller, Conceptualization, Data curation, Formal analysis, Funding acquisition, Investigation, Visualization, Methodology, Writing - original draft, Project administration, Writing – review and editing; Jeffrey M Lotthammer, Louis G Smith, Conceptualization, Data curation, Formal analysis, Funding acquisition, Investigation, Methodology, Project administration, Writing – review and editing; Borna Novak, Conceptualization, Data curation, Formal analysis, Investigation, Methodology, Writing – review and editing; Lindsey A Lee, Catherine C Kuhn, Data curation, Formal analysis, Investigation, Methodology, Writing – review and editing; Lina Greenberg, Data curation, Formal analysis, Investigation; Leslie A Leinwand, Data curation, Funding acquisition, Investigation, Writing – review and editing; Michael J Greenberg, Conceptualization, Funding acquisition, Investigation, Project administration, Writing – review and editing; Gregory R Bowman, Conceptualization, Funding acquisition, Investigation, Methodology, Project administration, Writing – review and editing

### Author ORCIDs

Artur Meller (ID) http://orcid.org/0000-0002-5504-2684
Jeffrey M Lotthammer (ID) http://orcid.org/0000-0002-5022-7006
Leslie A Leinwand (ID) http://orcid.org/0000-0003-1470-4810
Michael J Greenberg (ID) http://orcid.org/0000-0003-1320-3547
Gregory R Bowman (ID) http://orcid.org/0000-0002-2083-4892

### Decision letter and Author response

Decision letter https://doi.org/10.7554/eLife.83602.sa1
Author response https://doi.org/10.7554/eLife.83602.sa2

## Additional files

### Supplementary files
• MDAR checklist
• Supplementary file 1. The following supplementary data tables are available: IC50 values for

different myosin isoforms; Percent identity in motor domain sequence between myosin-II isoforms in this study; Structural similarity between myosin-II prepowerstroke state crystal structures as assessed by C-$\alpha$ root mean square deviation.

## Data availability

Experimental, pocket volume, docking, and trajectory clustering data have been deposited in OSF under accession code CV6D2. Scripts and notebooks used to generate all figures are available in our GitHub repository (https://github.com/bowman-lab/blebbistatin-specificity, copy archived at swh:1:rev:d0231df25d22780fde598cd4c35acca3b0d174d5).

The following dataset was generated:

| Author(s) | Year | Dataset title | Dataset URL | Database and Identifier |
|---|---|---|---|---|
| Meller A, Lotthammer JM, Smith LG, Novak B, Lee LA, Kuhn CK, Greenberg L, Leinwand LA, Greenberg MJ, Bowman GR | 2022 | class2-myosin-isoforms | https://osf.io/cv6d2/ | Open Science Framework, CV6D2 |

The following previously published dataset was used:

| Author(s) | Year | Dataset title | Dataset URL | Database and Identifier |
|---|---|---|---|---|
| Porter JR, Meller A, Zimmerman MI, Greenberg MJ, Bowman GR | 2019 | myosin-isoforms | https://osf.io/54g7p/ | Open Science Framework, 10.17605/OSF.IO/54G7P |

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
