## [Editor Report]

This study presents insights into how conformational dynamics differentially influences drug specificity and affinity in myosin isoforms using computational approaches. The evidence supporting the conclusions is convincing, establishing a relationship between inhibition and protein dynamics using state of the art computational techniques followed by experimental validation. The work is important and will be of broad interest to computational biophysicists and medicinal chemists.

---

## [Decision Letter]

**Decision letter after peer review:**

Thank you for submitting your article "Drug specificity and affinity are encoded in the probability of cryptic pocket opening in myosin motor domains" for consideration by *eLife*. Your article has been reviewed by 3 peer reviewers, and the evaluation has been overseen by Donald Hamelberg as Reviewing Editor and José Faraldo-Gómez as Senior Editor. All reviewers have opted to remain anonymous.

Essential revisions:

1) There is slight confusion on whether the authors are investigating specificity or affinity or both. Experiments so far have provided IC50s of blebbistatin, which the authors equivocate to Ki, i.e. affinity in their methods. However, the authors note in the discussion that specificity is usually related to composition of residues at the binding site, which is identical for most of the myosin-IIs studied here. Thus, it could be argued that blebbistatin binds all the myosins specifically. Are authors making the argument that a difference in binding affinity (again measured using IC50s as proxy) indicates a different in specificity, even if the pocket looks the same across isoforms? It is suggested that authors provide a brief discussion on their rationale here. Perhaps the specificity refers to the comparison between Myosin-IIs and other non-II isoforms, as opposed to between II-isoforms.

2) Homology models: It is evident that homology models are used for several of the simulations performed, but because there are so many states under investigation (ADP-bound, ATP-bound, nucleotide free, etc), it is not very clear exactly which structures are generated by homology model and the table provided in the Methods does not contain an exhaustive list. It would be helpful for authors to include a detailed table describing all proteins AND states simulated, and the source of these structures (i.e. Xray or homology). Additionally, it would be helpful to have more information about how appropriate the models are. What is the sequence identity/similarity between Myosins in this domain? For existing crystal structures of any 2 Myosin-II in the same state, what is their RMSD? This would assure readers of structural similarity in this protein family and the appropriateness of using models. Finally, how was stability of models assessed? If RMSDs were performed following simulations, these should be provided as assurance that the models were stable in simulation.

3) A major weakness of the work is the use of docking scores to compute the IC50 of blebbistatin for the different isoforms of myosin. Docking scores are usually empirical and previous works have shown that they are usually poorly correlated with experimental binding affinities. The work would be improved if the authors used a more physics-based approach with low computational cost to estimate the IC50 value, such as Molecular Mechanics Generalized-Boltzmann Surface Area (MM-GBSA).

4) On the correlation plot in Figure 6A, if myosins with IC50 > 150 mM were added to the plot, would the r^2^ correlation still be high?

5) In Figure 1 C and D, can authors label the residues they show in cyan to match the numbering in 1B? This would help readers make connections quicker.

6) For figure 4, can authors add the equilibrium probability (such as shown in Figure 3B) to each figure? This seems like a useful metric to quickly quantify probability of pocket opening from MSM.

7) Authors should clarify early on what is meant by 'apo' vs 'holo' structures. This becomes evident upon further reading, but not obvious early on as there is both an active site and allosteric site in this protein.

8) Can authors specify, of the 124 structures mined from the PDB used to check cryptic pocket existence, how many unique myosin sequences does this correspond to? What is distribution among families?

9) The authors show a figure of the open cryptic pocket observed in MD and how it overlays nicely (i.e with small RMSD) with holo crystal structure. Later, authors speculate that the aromatic side chain at position 466 points in the pocket and reduces the volume available for blebbistain binding. It would be interesting to see a representative figure from myosin V simulations in the predicted open conformation. Presumably blebbistatin also binds mysoin Vs, but just with lower affinity? Or is the IC50 of 150 mM in this case also taken as a proxy indicator that these myosins do not bind blebbistatin?

10) It could be interesting for authors to discuss on speculate on what may dictate pocket opening dynamics. Are these controlled by protein sequence elements? If so, why would they be so different for very similar isoforms?

11) It would be very interesting to read a discussion about the age-old concept of induced fit versus conformational selection. These data support conformational selection, but some information about types of sites for which cryptic pocket opening is partial or insufficient to support binding without ligand-induced changes would be appreciated.

12) It may have missed in the text, but does blebbistatin interfere with actin binding or the ability to exchange in ATP when bound to actin?

13) Greater clarification of the 'weighted' free energy from docking would be appreciated, particularly in light of the configurational entropy that may be inherent among the apo states of the protein.

14) Please clarify the simulation conditions for the existing dataset of apo motor fluctuations (line 246)

15) Long (>500 ns) simulations demonstrated that the cryptic side opens, but occurs late in the simulation. In light of which, would you have anticipated that a large ensemble of shorter simulations could uncover the same opening event from the same starting conditions? Similarly, could you speculate on whether enhanced sampling techniques (accelerated MD, meta dynamics etc) could achieve an equivalent outcome as was obtained by folding at home? A MSM based on shorter aMD simulations would strongly support your argument in this regard. If successful, this could make the MSM+docking approach appealing to the broader community that does not have ready access to the distributed computing framework.

16) Please clarify what is meant by conformation ensembles were enriched for 'such structures' (line 353)

17) The interpretation of the lag time data in the supplement is that the lifetimes are not fully converged with respect to lag time (skeletal and β cardiac) – is there any structural significance to those nonconvergent states?

18) π  $\pi$ (line 577 )

19) figure 1: letters in panels A and B have a light color, hard to read.

20) figure 2C: "Blebbistatin is shown in cyan with the pocket from the MD structure shown as a cyan contour."

It seems blebbistatin is in orange.

"Select residues…"

Typo, change to 'Selected'.

Residues Y269, L270, and F657 could be indicated in the figure to make it more informative.

Letters have light color, hard to read.

21) page 18: "When we docked to the closed apo experimental structures from which simulations were launched, we unsurprisingly found very poor docking scores for all myosin-II isoforms (Figure S6)."

Where was the bound pose located? In the cryptic pocket?

22) page 25, section "Preparation of homology models"

The authors could provide the sequence identity between the isoform modeled and the isoform in the PDB file, to help the readers evaluate whether homology modeling was reasonable for the cases presented.

23) table S1: units of the IC50 missing.

24) methods section: the authors could point the reader to important information available in table S2 (for instance, length of MD simulations and lag times chosen to build Markov state models).

---

## [Author Response]

Essential revisions:1) There is slight confusion on whether the authors are investigating specificity or affinity or both. Experiments so far have provided IC50s of blebbistatin, which the authors equivocate to Ki, i.e. affinity in their methods. However, the authors note in the discussion that specificity is usually related to composition of residues at the binding site, which is identical for most of the myosin-IIs studied here. Thus, it could be argued that blebbistatin binds all the myosins specifically. Are authors making the argument that a difference in binding affinity (again measured using IC50s as proxy) indicates a different in specificity, even if the pocket looks the same across isoforms? It is suggested that authors provide a brief discussion on their rationale here. Perhaps the specificity refers to the comparison between Myosin-IIs and other non-II isoforms, as opposed to between II-isoforms.

We would like to thank the reviewers for pointing out this ambiguity in language. We have taken the reviewers’ advice and narrowed the scope of “specificity.” Blebbistatin is referred to as a “myosin-II specific allosteric inhibitor,” and our manuscript addresses differences in *blebbistatin specificity* between myosin-IIs and non-myosin-IIs as well as *differences in affinity* between myosin-IIs. A few examples of the changes we have made are highlighted here:

The mechanism of blebbistatin’s specificity is not completely understood -> The mechanism by which blebbistatin differentially inhibits myosin isoforms is not completely understood.However, it is much less clear what the molecular determinants of blebbistatin specificity are within the myosin-II family. -> However, it is much less clear what molecular determinants explain differences in blebbistatin potency between isoforms in the myosin- II family.

2) Homology models: It is evident that homology models are used for several of the simulations performed, but because there are so many states under investigation (ADP-bound, ATP-bound, nucleotide free, etc), it is not very clear exactly which structures are generated by homology model and the table provided in the Methods does not contain an exhaustive list. It would be helpful for authors to include a detailed table describing all proteins AND states simulated, and the source of these structures (i.e. Xray or homology). Additionally, it would be helpful to have more information about how appropriate the models are. What is the sequence identity/similarity between Myosins in this domain? For existing crystal structures of any 2 Myosin-II in the same state, what is their RMSD? This would assure readers of structural similarity in this protein family and the appropriateness of using models. Finally, how was stability of models assessed? If RMSDs were performed following simulations, these should be provided as assurance that the models were stable in simulation.

We agree with the reviewers that homology modeling is an important part of our methods and greater details will help assure readers that homology modeling was performed appropriately.

Following the reviewer’s suggestion, we have added the nucleotide state for each target crystal structure to the table in the Methods that summarizes our homology modeling. We have also emphasized that starting structures for each our simulations were generated using homology modeling.

As requested by the reviewers, we have listed the sequence identity between the myosin sequence we modeled and the template crystal structure in a table in the Methods.

The high sequence similarity (>= 85% for all homology models) between input sequences and template structure sequences indicate that the selection of template structures was reasonable.

To further demonstrate that homology modeling between myosin-IIs is appropriate, we have also included information about the structural similarity between myosin-II crystal structures. We provide a table containing the C-α RMSD between the different prepowerstroke crystal structures used as templates for homology modeling in the supplement.

Finally, we rely on Markov State Models to address the possibility that input starting structures from homology modeling were low probability conformations in the ensemble of structures adopted by a given isoform. If the structures from homology modeling were high energy conformers, we would expect the simulation to transition away from these structures and the high flux away from these states to lead to low equilibrium probabilities in the final MSM.

3) A major weakness of the work is the use of docking scores to compute the IC50 of blebbistatin for the different isoforms of myosin. Docking scores are usually empirical and previous works have shown that they are usually poorly correlated with experimental binding affinities. The work would be improved if the authors used a more physics-based approach with low computational cost to estimate the IC50 value, such as Molecular Mechanics Generalized-Boltzmann Surface Area (MM-GBSA).

We agree that docking has significant limitations when it comes to predicting the relative affinities of different compounds for a target protein and share your interest in adopting more physically rigorous methods for such purposes. However, in this case, we reasoned that simple docking may capture the relative affinities of a single compound for different protein targets given that (1) the parameters are likely more reliable for the 20 amino acids than all of chemical space, given the greater availability of data, and (2) any errors in the description of the compound will be systematic. The correlation between previously measured IC50s and our predicted affinities is consistent with this hypothesis. More importantly, our experimental characterization of MYH7b shows that our approach was predictive in this case. Given this data, we believe that the inclusion of MM-GBSA calculations or other free energy methods is beyond the scope of this current work.

Based on the reviewer’s comments, we have included the following commentary on alternative approaches in the discussion (line 444):

Furthermore, we note that our analysis could be improved by using more rigorous, physics- based methods for predicting protein-ligand affinity (e.g., Molecular Mechanics Generalized Boltzmann Surface Area). Future work will investigate which sampling strategies and binding prediction methods are most efficient and accurate.

4) On the correlation plot in Figure 6A, if myosins with IC50 > 150 mM were added to the plot, would the r^2^ correlation still be high?

We would like to thank the reviewers for this suggestion. It is something we have considered doing. However, blebbistatin’s affinity for non-myosin-IIs is beyond the detectable limit from experiments (blebbistatin’s limited solubility prevents testing at high concentration). Hence, it is not clear what value to use for the experimental affinity of these isoforms, so we do not believe it is possible to include them in the *R*^!^ calculation.

5) In Figure 1 C and D, can authors label the residues they show in cyan to match the numbering in 1B? This would help readers make connections quicker.

We would like to thank the reviewers for this suggestion. We have labeled the residues displayed in figure 1C and 1D with the same numbering as in 1B. This will allow readers to better understand sequences differences at the blebbistatin pocket.

6) For figure 4, can authors add the equilibrium probability (such as shown in Figure 3B) to each figure? This seems like a useful metric to quickly quantify probability of pocket opening from MSM.

The equilibrium probabilities for each isoform are included in a supplemental figure. Since this may have been easy to miss in our previous version, we have included explicit mention of this figure in the caption.

7) Authors should clarify early on what is meant by 'apo' vs 'holo' structures. This becomes evident upon further reading, but not obvious early on as there is both an active site and allosteric site in this protein.

We would like to thank the reviewers for pointing out this ambiguity. We have replaced all instances of “apo” with “blebbistatin-free” to clarify that these structures did not include blebbistatin. We define “holo” explicitly on line 147 and list the reference PDB ID for further clarification when it was reasonable to do so.

8) Can authors specify, of the 124 structures mined from the PDB used to check cryptic pocket existence, how many unique myosin sequences does this correspond to? What is distribution among families?

There were 23 unique sequences represented in the 124 myosin structures mined from the PDB. Of these 23 sequences, 11 were from the myosin-II family while the other 12 were from other myosin families (e.g., myosin-Vs, myosin-Xs, etc.). We have added these details to the Methods to demonstrate that this set of structures comprised a diverse collection of myosin isoforms.

9) The authors show a figure of the open cryptic pocket observed in MD and how it overlays nicely (i.e with small RMSD) with holo crystal structure. Later, authors speculate that the aromatic side chain at position 466 points in the pocket and reduces the volume available for blebbistain binding. It would be interesting to see a representative figure from myosin V simulations in the predicted open conformation. Presumably blebbistatin also binds mysoin Vs, but just with lower affinity? Or is the IC50 of 150 mM in this case also taken as a proxy indicator that these myosins do not bind blebbistatin?

As the reviewer observes, the IC50 listed for myosin-V (> 150 mM) is meant to indicate that blebbistatin does not bind this isoform within the detectable limit of experiments. In simulations of myosin-V, the blebbistatin pocket remains closed, never reaching the *holo* volume. Indeed, myosin-V has a striking lack of conformational heterogeneity in the position of its U50 linker, see Author response image 1.

**Author response image 1. sa2fig1:** 

The green structure is the holo crystal structure; the cyan structure is the blebbistatin-free myosin-V crystal structure; and the gray residues indicate the positions of the leucine in the U50 linker for myosin-V across a subset of MSM cluster centers (this leucine never adopts its blebbistatin-bound orientation in myosin-V simulations).

10) It could be interesting for authors to discuss on speculate on what may dictate pocket opening dynamics. Are these controlled by protein sequence elements? If so, why would they be so different for very similar isoforms?

The reviewers pose very interesting questions. We have added the following paragraph to address these points (line 430):

“Given there are differences in pocket opening probabilities even when pocket-lining residues are identical or near identical, distant sequence elements must allosterically modulate pocket opening. Our results demonstrate that physics-based simulations can capture these allosteric networks by modeling complex couplings throughout the myosin molecule. Even though myosin motors from the myosin-II family share high sequence identity (Table S3) and structural similarity, they are evolutionarily tuned to explore different distributions of conformations in solution.^40^ Future work is needed to isolate precisely which sequence elements control blebbistatin pocket opening.”

11) It would be very interesting to read a discussion about the age-old concept of induced fit versus conformational selection. These data support conformational selection, but some information about types of sites for which cryptic pocket opening is partial or insufficient to support binding without ligand-induced changes would be appreciated.

This is an excellent suggestion, and this addition will undoubtedly strengthen our Discussion section. We have added the following paragraph discussing conformational selection vs. induced fit (line 458):

“Additionally, our finding that the blebbistatin pocket readily opens in simulation is consistent with an important role for conformational selection. Many other simulation studies have also demonstrated that protein conformational changes associated with ligand binding are present in the ligand-free ensemble of structures adopted by proteins in solution. Furthermore, a growing body of experimental evidence supports a primary role for conformational selection. However, there are examples from our previous work (e.g., Niemann-Pick Protein C2) where even several rounds of adaptive sampling simulations do not sample cryptic pocket opening. These results along with a flux-based analysis of binding mechanisms suggest that induced fit may also play an important role, especially at high ligand concentrations.”

12) It may have missed in the text, but does blebbistatin interfere with actin binding or the ability to exchange in ATP when bound to actin?

We thank the reviewer for raising the question. Light scattering and pyrene-actin fluorescence measurements as well as ultracentrifugation experiments have shown that blebbistatin interferes with actin binding. We have added this point to our introduction. Blebbistatin does not have a significant effect on ATP-induced dissociation of actomyosin most likely because it cannot bind myosin when it is bound to actin. We have added the following line about blebbistatin’s mechanism of inhibition (line 80):

“Blebbistatin inhibits myosin ATPase by preventing the release of phosphate from the active site and interfering with actin binding.”

13) Greater clarification of the 'weighted' free energy from docking would be appreciated, particularly in light of the configurational entropy that may be inherent among the apo states of the protein.

We would like to thank the reviewers for providing this opportunity to clarify how our MSM- docking approach considers protein configurational entropy. In contrast to docking to a single structure, we directly consider a protein’s configurational entropy by docking to a diverse ensemble of structures from the apo simulations. The overall free energy of binding from docking to an MSM can be written as:ΔGtotal= −kBTln\ (∑iπiKeqi) where *π* is the equilibrium probability of the MSM state and Keqi is the equilibrium association constant for a MSM microstate calculated from the docking score for that MSM state.

If a protein has nearly all closed states, then most of its docking scores will be unfavorable. These states will minimally contribute to the macro-association constant, which is the weighted sum of micro-association constants. Similarly, states that are low probability will have a small contribution to the overall macro-association constant. Conversely, if a protein has mostly open states, those states will receive favorable docking scores and contribute substantially to the macro-association constant. Hence, the MSM-docking approach directly considers the distribution of open and closed states in the apo ensemble.

We have clarified this argument in the Methods section (line 623):

“We note that this formula directly considers the configurational entropy of the protein by taking a weighted average of the micro-association constants over all the states in the *apo* MSM. Closed states will receive unfavorable (i.e., more positive) docking scores reflecting in small association constants while open states will receive favorable (i.e., more negative) docking scores resulting in large association constants. Similarly, high probability states will contribute more to the weighted sum of micro-association constants, which is the macro-association constant of the ligand binding reaction. Hence, a protein that pays a high entropic penalty for ligand binding because it has only a few, low probability open states will have a small macro- association constant and an unfavorable overall MSM-docking score. Conversely, a protein that pays a low entropic penalty for ligand binding because of its many, high probability open conformers will have a large macro-association constant and hence a more favorable overall MSM-docking score.”

14) Please clarify the simulation conditions for the existing dataset of apo motor fluctuations (line 246)

In the main text, we have clarified that these simulations were conducted without blebbistatin and without nucleotide in the active site. We have also provided additional details about this dataset in the Methods and refer the reader to the Methods in Porter *et. al* for a full explanation.

15) Long (>500 ns) simulations demonstrated that the cryptic side opens, but occurs late in the simulation. In light of which, would you have anticipated that a large ensemble of shorter simulations could uncover the same opening event from the same starting conditions? Similarly, could you speculate on whether enhanced sampling techniques (accelerated MD, meta dynamics etc) could achieve an equivalent outcome as was obtained by folding at home? A MSM based on shorter aMD simulations would strongly support your argument in this regard. If successful, this could make the MSM+docking approach appealing to the broader community that does not have ready access to the distributed computing framework.

The reviewers raise an excellent point. As suggested by the reviewer, we would expect to uncover the same opening event from a large ensemble of shorter simulations. Indeed, this is what we see in our folding@home simulations launched from a closed state.

The reviewer also points out that our MSM-docking approach may be more broadly interesting if it does not require exhaustive sampling via distributed computing. Based on our other work assessing cryptic pocket opening across 12 systems, we would speculate that enhanced sampling techniques or even 10 relatively short MD simulations would be sufficient to sample many cryptic pocket opening events. We have added these details to a new paragraph (line 438):

“Though our present work relies on combining long simulations and highly parallel simulations gathered on folding@home with docking, we speculate that several alternative methods could produce similar results. Efficient sampling of cryptic pocket opening is possible with adaptive sampling methods like FAST or enhanced sampling methods like metadynamics and accelerated molecular dynamics. In many cases, a relatively modest amount of sampling is sufficient to observe cryptic pocket opening, though accurate estimation of opening probabilities may require additional sampling.”

16) Please clarify what is meant by conformation ensembles were enriched for 'such structures' (line 353)

We agree with the reviewer that this particular line was ambiguous. We have rewritten this sentence as follows (line 355):

“All four myosin-II isoforms had docked poses with ligand heavy atom RMSD < 3 Å from the *holo* pose, but skeletal muscle and β-cardiac myosin have a substantially higher probability of adopting conformations where blebbistatin docks in *holo*-like poses (Figure S7).”

17) The interpretation of the lag time data in the supplement is that the lifetimes are not fully converged with respect to lag time (skeletal and β cardiac) – is there any structural significance to those nonconvergent states?

We do not believe that there is any significance to the minor fluctuations in implied timescales seen in these plots.

18) π  $\pi$ (line 577 )

We have made this change.

19) figure 1: letters in panels A and B have a light color, hard to read.

We have darkened the colors in those panels to make them more readable.

20) figure 2C: "Blebbistatin is shown in cyan with the pocket from the MD structure shown as a cyan contour."It seems blebbistatin is in orange."Select residues…"Typo, change to 'Selected'.Residues Y269, L270, and F657 could be indicated in the figure to make it more informative.Letters have light color, hard to read.

We appreciate these suggestions and have updated figure 2 as suggested.

21) page 18: "When we docked to the closed apo experimental structures from which simulations were launched, we unsurprisingly found very poor docking scores for all myosin-II isoforms (Figure S6)."Where was the bound pose located? In the cryptic pocket?

The bound poses for these experimental structures were found outside the cryptic pocket in an adjacent cavity. The low quality of these poses as assessed by docking scores suggests non- specific binding to these protein conformations.

To clarify this point, we have revised this section as follows (line 337):

“When we docked blebbistatin to the closed experimental structures from which simulations were launched, we found that it occupied an adjacent cavity but that these poses received very poor docking scores for all myosin-II isoforms (Figure S6). There were no subtle structural differences between experimental structures or adjacent pockets that might explain variation in blebbistatin potency.”

22) page 25, section "Preparation of homology models"The authors could provide the sequence identity between the isoform modeled and the isoform in the PDB file, to help the readers evaluate whether homology modeling was reasonable for the cases presented.

This is a very good suggestion from the reviewers. We have done exactly this and provided these details in the Methods under “Preparation of homology models.”

23) table S1: units of the IC50 missing.

We have added units to this table.

24) methods section: the authors could point the reader to important information available in table S2 (for instance, length of MD simulations and lag times chosen to build Markov state models).

We thank the reviewers for this suggestion. Since these are important parameters, we have moved this table to the main text.